# A Semi-Bayesian Nonparametric Estimator of the Maximum Mean Discrepancy Measure: Applications in Goodness-of-Fit Testing and Generative Adversarial Networks

**Forough Fazeli-Asl**                                                          *foroughf@hku.hk*
*Department of Statistics and Actuarial Science*
*University of Hong Kong*

**Michael Minyi Zhang**                                                         *mzhang18@hku.hk*
*Department of Statistics and Actuarial Science*
*University of Hong Kong*

**Lizhen Lin**                                                                  *lizhen01@umd.edu*
*Department of Mathematics*
*The University of Maryland*
*College Park, MD, USA*

**Reviewed on OpenReview:** *https://openreview.net/forum?id=lUnlHS1FYT*

## Abstract

A classic inferential problem in statistics is the goodness-of-fit (GOF) test. Performing such tests can be challenging when the hypothesized parametric model has an intractable likelihood and its distributional form is not available. Bayesian methods for GOF testing can be appealing due to their ability to incorporate expert knowledge through prior distributions. However, standard Bayesian methods for this test often require strong distributional assumptions on the data and their relevant parameters. To address this issue, we propose a semi-Bayesian nonparametric (semi-BNP) procedure based on the maximum mean discrepancy (MMD) measure that can be applied to the GOF test. We introduce a novel Bayesian estimator for the MMD, which enables the development of a measure-based hypothesis test for intractable models. Through extensive experiments, we demonstrate that our proposed test outperforms frequentist MMD-based methods by achieving a lower false rejection and acceptance rate of the null hypothesis. Furthermore, we showcase the versatility of our approach by embedding the proposed estimator within a generative adversarial network (GAN) framework. It facilitates a robust BNP learning approach as another significant application of our method. With our BNP procedure, this new GAN approach can enhance sample diversity and improve inferential accuracy compared to traditional techniques.

## 1 Introduction

Goodness-of-fit (GOF) tests are commonly used to evaluate an empirical data set against a hypothesized parametric model. However, there are cases when the likelihood of the parametric model is intractable and the explicit form of the model distribution is unavailable, making it challenging to directly assess the model's fit. One such example is the case of deep generative models, where independent samples can be generated, but the required likelihood function needed for traditional GOF tests is intractable. In such situations, a potential solution is to use the the maximum mean discrepancy (MMD) measure as an alternative approach for conducting GOF tests (Gretton et al., 2012a; Key et al., 2021) in addition to some existing kernelized methods (Liu et al., 2016). The MMD is a metric on the space of probability distributions and is commonly used in hypothesis testing to quantify the difference between the distribution of the data and the hypothesized model.

It can be conveniently estimated using available samples generated from desired distributions. The MMD estimator has proven to be effective in various applications, including analyzing large-scale datasets with high-dimensional features and implementing generative models, especially generative adversarial networks (GANs).

Bayesian nonparametric methods, while powerful, have received comparatively little attention, especially regarding their application in estimating the MMD. One of the primary benefits of the Bayesian approach is that expert knowledge can be incorporated into the prior distributions in a diagnostic setting. Moreover, a BNP learning procedure can provide a certain level of regularization to the training process. This is partially a result of placing uncertainty on the sampling distribution of the data, via a Dirichlet process (DP). Therefore, the lack of such methods in MMD estimation proves to be a hindrance for the statistician who wishes to be Bayesian without overly strong assumptions. This paper seeks to fill this crucial gap.

In this paper, we propose a BNP estimator that accurately estimates the MMD kernel-based measure between an intractable parametric model and an unknown distribution. To develop the procedure, we place the DP prior solely on the unknown distribution. Therefore, we refer to this procedure as a semi-Bayesian nonparametric (semi-BNP) estimator. Having established our MMD estimator, we demonstrate that we can generalize the bootstrap procedure given in Dellaporta et al. (2022) beyond posterior parameter inference. First, we apply our estimator in a variety of hypothesis testing problems. Next, we introduce a robust Bayesian nonparametric learning (BNPL) approach for training GANs based on simulating from the posterior distribution on the parameter space of the generator. Our approach utilizes the aforementioned estimator as a robust discriminator between the generator's distribution and a DP posterior on the empirical data distribution. Specifically, our framework unifies concepts of the MMD measurement and the BNP inference to leverage their respective benefits into a single discriminator. Furthermore, we will investigate the ability of our discriminator to reduce mode collapse and increase the ability of the generator to fool the discriminator more effectively than the frequentist counterpart for GAN training.

The paper is organized as follows: In Section 2, we review previous works and methods related to our proposed technique. We then introduce our novel semi-BNP estimator for the MMD measure between an unknown and intractable parametric distribution in Section 3, and provide theoretical properties of our proposed estimator. In Section 4, we utilize our semi-BNP estimator of the MMD measure to create a powerful GOF test based on the relative belief (RB) ratio, which serves as the Bayesian evidence to judge the null hypothesis. Moreover, Section 5 outlines the incorporation of the semi-BNP estimator as the discriminator in the GAN architecture. This results in a robust BNPL procedure that accurately estimates the generator's parameters for generating realistic samples. The section also discusses the theoretical properties of the proposed discriminator, such as robustness and consistency. We evaluate the novel semi-BNP procedures for hypothesis testing and GAN training through numerical experiments in Section 6. Lastly, we conclude the paper in Section 7 and discuss potential future directions. All proofs, algorithms, notations, and additional experiments are given in the Appendix.

## 2 Previous Work

Our proposed method consists of two fundamental components: the MMD measure and the DP prior. First, we will review these two concepts.

### 2.1 Maximum Mean Discrepancy Measure

For a given data space $\mathfrak{X}$, consider the random variables $\mathbf{X}$ and $\mathbf{Y}$, drawn from distributions $F_1$ and $F_2$ respectively. Here, $F_1$ and $F_2$ belong to $\mathcal{B}(\mathfrak{X})$, which represents the set of Borel probability distributions on $\mathfrak{X}$. We consider the discrepancy $d : \mathcal{B}(\mathfrak{X}) \times \mathcal{B}(\mathfrak{X}) \to [0, \infty)$ through the integral pseudo-probability metric (IPM) (Müller, 1997), defined as shown in (1). The class of functions $\mathcal{F}$ is designed to be rich enough to distinguish between $F_1$ and $F_2$, and restrictive enough to provide accurate estimates based on a finite sample.

$$d_{\mathrm{IPM}}(F_1, F_2) = \sup_{h \in \mathcal{F}} |E_{F_1}(h(\mathbf{X})) - E_{F_2}(h(\mathbf{Y})))|. \tag{1}$$

The MMD is then defined by considering $\mathcal{F} = \{h \in \mathcal{H}_k | \, ||h||_{\mathcal{H}_k} \leq 1\}$, which represents a unit ball in a reproducing kernel Hilbert space (RKHS) $\mathcal{H}_k$ with associated kernel $k : \mathfrak{X} \times \mathfrak{X} \to \mathbb{R}$. In this context, $||\cdot||_{\mathcal{H}_k}$ denotes the norm function in the RKHS. The function $k(\cdot, \cdot)$ is positive definite, such that for any function $h \in \mathcal{H}_k$ and any $\mathbf{X} \in \mathfrak{X}$, $h(\mathbf{X}) = \langle h, k(\mathbf{X}, \cdot) \rangle_{\mathcal{H}_k}$, where $\langle \cdot, \cdot \rangle_{\mathcal{H}_k}$ represents the inner product in $\mathcal{H}_k$. Consider function $\mu_{F_1}(\cdot) = E_{F_1}[k(\mathbf{X}, \cdot)] \in \mathcal{H}_k$, which is defined as the kernel mean embedding of the distribution $F_1$ in Gretton et al. (2012a). Then, for given $\mathbf{X}, \mathbf{X}' \overset{i.i.d.}{\sim} F_1, \mathbf{Y}, \mathbf{Y}' \overset{i.i.d.}{\sim} F_2$, if $E_F(\sqrt{k(\mathbf{X}, \mathbf{X})}) < \infty$ for all $F \in \mathcal{B}(\mathfrak{X})$, the MMD is given by

$$\mathrm{MMD}^2(F_1, F_2) = ||\mu_{F_1} - \mu_{F_2}||^2_{\mathcal{H}_k} = E_{F_1}[k(\mathbf{X}, \mathbf{X}')] - 2E_{F_1, F_2}[k(\mathbf{X}, \mathbf{Y})] + E_{F_2}[k(\mathbf{Y}, \mathbf{Y}')]. \tag{2}$$

Note that $\mathrm{MMD}^2(F_1, F_2) = 0$ if and only if $F_1 = F_2$, when $\mathcal{H}_k$ is a *universal* RKHS defined on a *compact* metric space $\mathfrak{X}$ and $k(\cdot, \cdot)$ is *continuous* (Gretton et al., 2012a, Theorem 5). In practice, distributions $F_1$ and $F_2$ are not accessible, and then the biased, empirical estimator of (2) (V-statistic) is calculated using empirical distributions $F_{1,n}$ and $F_{2,m}$ as

$$\mathrm{MMD}^2(F_{1,n}, F_{2,m}) = \frac{1}{n^2} \sum_{i,j=1}^{n} k(\mathbf{X}_i, \mathbf{X}_j) - \frac{2}{mn} \sum_{i=1}^{n} \sum_{j=1}^{m} k(\mathbf{X}_i, \mathbf{Y}_j) + \frac{1}{m^2} \sum_{i,j=1}^{m} k(\mathbf{Y}_i, \mathbf{Y}_j), \tag{3}$$

where $\mathbf{X}_1, \ldots, \mathbf{X}_n$ is a sample from $F_1$ and $\mathbf{Y}_1, \ldots, \mathbf{Y}_m$ is a sample generated from $F_2$.

Recently, Key et al. (2021) proposed a GOF test using the MMD measure when the hypothesized model belongs to a parametric family of intractable models. It was proposed to be employed in training generative models such as toggle-switch models and GANs. There are also numerous generative models closely linked to the implementation of MMD in GANs, which can be found in Briol et al. (2019), Niu et al. (2023), Oates (2022), and Bharti et al. (2023). These models offer distinct MMD estimators that are specifically designed to further improve the MMD's capability in estimating the generator's parameters.

## 2.2 Bayesian Methods: Approximate Bayesian Computation, the Dirichlet Process and Bayesian Nonparametric Learning

Previous work in simulation-based inference has largely focused on applying discrepancy measures from a frequentist nonparametric (FNP) perspective. A Bayesian perspective on simulation-based inference involves a similar methodology, using approximate Bayesian computation (ABC) to estimate the model parameters via simulation (Beaumont et al., 2002). In ABC, we first sample a proposal by sampling from a prior distribution placed on the parameter space of the generative model (Step 1). Rather than inferring parameters directly from the posterior distribution, we compare the summary statistics of the simulated data, given the current state of the parameters, with those of the observed data using a discrepancy measure (Step 2). The simulated parameter values corresponding to the accepted summary statistics are retained if the distance falls within a predetermined threshold (Step 3).

Identifying informative summary statistics in ABC is a challenging task and it depends on the specific application of the data being analyzed, such as the mean effective heterozygosity or the mean of variance in repeat numbers in genetic populations (Blum & François, 2010; Csilléry et al., 2012). An inappropriate choice may result in poor posterior inference from the data (Robert et al., 2011; Aeschbacher et al., 2012). One solution proposed by Park et al. (2016) is to use the MMD metric between simulated and real data distributions to avoid manually selecting the summary statistics. However, as the threshold approaches zero, ABC tends to the standard Bayesian posterior, which is susceptible to model misspecification and lacks robustness (Dellaporta et al., 2022)[1]. To address these two issues, generalized Bayesian inference (GBI) proposes an alternative method by replacing the likelihood in the posterior distribution with the exponential of a robust loss function.

For a detailed exploration, Jewson et al. (2018) offers various examples, including the use of exponential of the Hellinger divergence within the GBI. Within the GBI framework, there are two prominent procedures that use

---

[1]A detailed discussion is given in (Dellaporta et al., 2022, Sections 2); however, we have provided some details in the Appendix for easier access.

the MMD loss. Chérief-Abdellatif & Alquier (2020) propose a pseudo-likelihood based on the MMD metric and approximate the posterior using variational inference. Pacchiardi & Dutta (2021) extend this method to a more general Bayesian likelihood-free model using stochastic gradient Markov chain Monte Carlo (MCMC) to perform posterior inference[2]. Stein variational gradient descent (Liu & Wang, 2016) is another variational inference method that uses gradient-based updates derived from Stein's method and kernel functions. It leverages Bayesian principles to iteratively move particles to match the posterior distribution by minimizing Kullback-Leibler divergence, particularly when exact inference is computationally intractable (Feng et al., 2017).

However, Dellaporta et al. (2022) noted that the performance of GBI is very sensitive to the choice of a learning rate and that there is no general heuristic for selecting this hyperparameter. Additionally, these calculations often require MCMC sampling methods, which can impose a significant computational burden. To address these issues, Dellaporta et al. (2022) developed an MMD posterior bootstrap procedure following the BNPL strategy developed in Lyddon et al. (2018; 2019); Fong et al. (2019). In this BNPL strategy, a BNP prior is defined on $F$, leading to a BNP posterior on $F$, denoted by $F^{pos}$. The key idea is that any posterior on the generator's parameter space $\mathcal{W}$ can be derived by mapping $F^{pos}$ through the push-forward measure

$$\boldsymbol{\omega}^*(F^{pos}) := \arg\min_{\boldsymbol{\omega} \in \mathcal{W}} \delta(F^{pos}, F_{G_{\boldsymbol{\omega}}}),$$

which is visually depicted in Dellaporta et al. (2022, Figure 1). In particular, Dellaporta et al. (2022) considered $F^{pos}$ as the DP posterior and $\delta$ as the MMD measure.

The DP, introduced by Ferguson (1973), is a commonly used prior in Bayesian nonparametric methods. It can be viewed as an infinite-dimensional generalization of the Dirichlet distribution constructed around $H$ (the base measure), a fixed probability measure, whose variation is controlled by $a$ (the concentration parameter), a positive real number. To formally define the DP, consider a space $\mathfrak{X}$ with a $\sigma$-algebra $\mathcal{A}$ of subsets of $\mathfrak{X}$. For a base measure $G$ on $(\mathfrak{X}, \mathcal{A})$ and $a > 0$, a random probability measure $F = \{F(A) : A \in \mathcal{A}\}$ is called a DP on $(\mathfrak{X}, \mathcal{A})$, denoted by $F^{pri} := (F \sim DP(a, H))$, if for every measurable partition $A_1, \ldots, A_k$ of $\mathfrak{X}$ with $k \geq 2$, the joint distribution of the vector $(F(A_1), \ldots, F(A_k))$ has the Dirichlet distribution with parameters $(aH(A_1), \ldots, aH(A_k))$. It is assumed that $H(A_j) = 0$ implies $F(A_j) = 0$ with probability one.

One of the most important properties of the DP is the conjugacy property–when the sample $\mathbf{X}_{1:n} = (\mathbf{X}_1, \ldots, \mathbf{X}_n)$ is drawn from $F \sim DP(a, H)$, the posterior distribution of $F$ given $\mathbf{X}_{1:n}$, denoted by $F^{pos}$, is also a DP with concentration parameter $a + n$ and base measure

$$H^* = a(a+n)^{-1}H + n(a+n)^{-1}F_n,$$

where $F_n$ denotes the empirical cumulative distribution function (ECDF) of the sample $\mathbf{X}_{1:n}$. Note that, $H^*$ is a convex combination of the base measure $H$ and $F_n$. A guideline for choosing the hyperparameters $a$ and $H$ for the test of equality distributions will be covered in Section 4.

In previous work, there are several BNP GOF tests (Al-Labadi & Evans, 2018; Al-Labadi et al., 2021a;b), as well as two-sample tests (Al-Labadi & Zarepour, 2017; Al-Labadi, 2021) and a multi-sample test (Al-Labadi et al., 2022a), that are closely connected to the posterior-based distance estimation employed in the BNPL procedure of Dellaporta et al. (2022). These methods are developed using different discrepancy measures to compare the distance between DP posteriors, placed on unknown distributions, with the corresponding one between DP priors. However, unlike our proposed method, none of them employ the MMD measure.

Sethuraman (1994) proposed an infinite series representation as an alternative definition for DP. The construction of Sethuraman (1994) is known as the stick-breaking representation and is a popularly used method in DP inference. Particularly, for a sequence of identically distributed (i.i.d.) random variables $\{\beta_i\}_{i \geq 1}$ from Beta$(1, a)$, let $w_1 = \beta_1$, and $w_i = \beta_i \prod_{j=1}^{i-1}(1 - \beta_j)$, for $i \geq 2$. Then, the stick-breaking representation is given by $F_{SB} = \sum_{i=1}^{\infty} w_i \delta_{Y_i}$, where $\{Y_i\}_{i \geq 1}$ is a sequence of i.i.d. random variables from $H$. However, Zarepour & Al-Labadi (2012) addressed some difficulties in using these representations. Meanwhile, Ishwaran &

---

[2]A comprehensive list of other GBI procedures for addressing this issue can be found in Dellaporta et al. (2022).

Zarepour (2002) proposed a finite representation to facilitate the simulation of the DP. Let

$$F_N^{pri} = \sum_{i=1}^{N} J_{i,N} \delta_{Y_i},$$

where $(J_{1,N}, \ldots, J_{N,N}) \sim \text{Dirichlet}(a/N, \ldots, a/N)$, and $Y_i \overset{i.i.d.}{\sim} H$. Ishwaran & Zarepour (2002) showed that $\{F_N\}_{N=1}^{\infty}$ converges in distribution to $F$, where $F_N$ and $F$ are random values in the space $M_1(\mathbb{R})$ of probability measures on $\mathbb{R}$ endowed with the topology of weak convergence. Thus, to generate $\{J_{i,N}\}_{i=1}^{N}$ put $J_{i,N} = \Gamma_{i,N} / \sum_{i=1}^{N} \Gamma_{i,N}$, where $\{\Gamma_{i,N}\}_{i=1}^{N}$ is a sequence of i.i.d. $\text{Gamma}(a/N, 1)$ random variables independent of $\{Y_i\}_{i=1}^{N}$. This form of approximation leads to some results in subsequent sections.

To determine the number of DP approximation terms, we apply a random stopping rule, inspired by the method described in Zarepour & Al-Labadi (2012). This rule, given a specific $\epsilon \in (0, 1)$, is defined as:

$$N = \inf \left\{ j : \frac{\Gamma_{j,j}}{\sum_{i=1}^{j} \Gamma_{i,j}} < \epsilon \right\}. \tag{4}$$

## 3  A Semi-BNP MMD Estimator

This section introduces our semi-BNP estimator for approximating the MMD measure. We consider a scenario where $F_1$ represents a completely unknown distribution, while $F_2$ represents an intractable parametric distribution with a complex generating process. For a given sample $\mathbf{Y}_1, \ldots, \mathbf{Y}_m$ from $F_2$ and by assuming $F_1^{pri} := (F_1 \sim DP(a, H))$ for a non-negative value $a$ and a fixed probability measure $H$, we propose the prior-based MMD estimator as

$$\text{MMD}_{\text{BNP}}^2(F_{1,N}^{pri}, F_{2,m}) = \sum_{\ell,t=1}^{N} J_{\ell,N} J_{t,N} k(\mathbf{V}_\ell, \mathbf{V}_t) - \frac{2}{m} \sum_{\ell=1}^{N} \sum_{t=1}^{m} J_{\ell,N} k(\mathbf{V}_\ell, \mathbf{Y}_t) + \frac{1}{m^2} \sum_{\ell,t=1}^{m} k(\mathbf{Y}_\ell, \mathbf{Y}_t), \tag{5}$$

where $(J_{1,N}, \ldots, J_{N,N})$ is sampled from $\text{Dirichlet}(a/N, \ldots, a/N)$, $\mathbf{V}_1, \ldots, \mathbf{V}_N \overset{i.i.d.}{\sim} H$, and $N$ is the number of terms in the DP approximation $\sum_{\ell=1}^{N} J_{\ell,N} \delta_{\mathbf{V}_\ell}$ proposed by Ishwaran & Zarepour (2002). Since we only impose the DP prior on the distribution of the real data, we refer to the approach as a semi-BNP procedure.

**Theorem 1** *For a non-negative real value $a$ and fixed probability distribution $H$, let $F_1^{pri} := (F_1 \sim DP(a, H))$, $H_N$ be the ECDF corresponding to $H$, and $k(\cdot, \cdot)$ be any continuous kernel function with feature space corresponding to a universal RKHS defined on a compact metric space $\mathfrak{X}$. Assume that $|k(\boldsymbol{z}, \boldsymbol{z}')| < K$, for any $\boldsymbol{z}, \boldsymbol{z}' \in \mathbb{R}^d$. Then,*
*i. $\text{MMD}_{\text{BNP}}^2(F_{1,N}^{pri}, F_{2,m}) \xrightarrow{a.s.} \text{MMD}^2(H_N, F_{2,m})$, as $a \to \infty$,*
*ii. $E(\text{MMD}_{\text{BNP}}^2(F_{1,N}^{pri}, F_{2,m})) \to \text{MMD}^2(H, F_2)$ as $a \to \infty$, $N \to \infty$, and $m \to \infty$,*
*iii. $E(\text{MMD}_{\text{BNP}}^2(F_{1,N}^{pri}, F_{2,m})) < \text{MMD}^2(H, F_2) + 3K$, for any $N, m \in \mathbb{N}$ and $a \in \mathbb{R}^+$,*
*where "$\xrightarrow{a.s.}$" denotes the almost surely convergence, $\mathbb{N}$ denotes the natural numbers and $\mathbb{R}^+$ denotes the positive real numbers.*

After observing samples $\mathbf{X}_1, \ldots, \mathbf{X}_n$ from $F_1$ and considering $\mathbf{V}_1^*, \ldots, \mathbf{V}_N^* \overset{i.i.d.}{\sim} H^*$, and $(J_{1,N}^*, \ldots, J_{N,N}^*) \sim \text{Dirichlet}(\frac{a+n}{N}, \ldots, \frac{a+n}{N})$, we update the prior-based MMD estimator (5) to the posterior one as

$$\text{MMD}_{\text{BNP}}^2(F_{1,N}^{pos}, F_{2,m}) = \sum_{\ell,t=1}^{N} J_{\ell,N}^* J_{t,N}^* k(\mathbf{V}_\ell^*, \mathbf{V}_t^*) - \frac{2}{m} \sum_{\ell=1}^{N} \sum_{t=1}^{m} J_{\ell,N}^* k(\mathbf{V}_\ell^*, \mathbf{Y}_t) + \frac{1}{m^2} \sum_{\ell,t=1}^{m} k(\mathbf{Y}_\ell, \mathbf{Y}_t), \tag{6}$$

where, $H^* = a/(a+n)H + n/(a+n)F_{1,n}$, $F_{1,n}$ denotes the empirical distribution of observed data, and $F_{1,N}^{pos}$ refers to the approximation of $F_1 | \mathbf{X}_{1:n} \sim DP(a+n, H^*)$. The following Theorem presents asymptotic properties of $\text{MMD}_{\text{BNP}}^2(F_{1,N}^{pos}, F_{2,m})$.

**Theorem 2** *For a non-negative real value $a$ and fixed probability distribution $H$, let $F_1^{pri} := (F_1 \sim DP(a, H))$ and $k(\cdot, \cdot)$ be any continuous kernel function with feature space corresponding to a universal RKHS defined on a compact metric space $\mathfrak{X}$. Assume that $|k(\boldsymbol{z}, \boldsymbol{z}')| < K$, for any $\boldsymbol{z}, \boldsymbol{z}' \in \mathbb{R}^d$. Then, for a given sample $\mathbf{X}_1, \ldots, \mathbf{X}_n$ from distribution $F_1$,*
*i. as $a \to \infty$ (informative prior),*

    *a.* $\mathrm{MMD}^2_{\mathrm{BNP}}(F_{1,N}^{pos}, F_{2,m}) \xrightarrow{a.s.} \mathrm{MMD}^2(H_N, F_{2,m}),$

    *b.* $E(\mathrm{MMD}^2_{\mathrm{BNP}}(F_{1,N}^{pos}, F_{2,m})) \to \mathrm{MMD}^2(H, F_2), N \to \infty,$ *and* $m \to \infty,$

*ii. as $n \to \infty$ (consistency),*

    *a.* $\mathrm{MMD}^2_{\mathrm{BNP}}(F_{1,N}^{pos}, F_{2,m}) \xrightarrow{a.s.} \mathrm{MMD}^2(F_{1,N}, F_{2,m}),$

    *b.* $E(\mathrm{MMD}^2_{\mathrm{BNP}}(F_{1,N}^{pos}, F_{2,m})) \to \mathrm{MMD}^2(F_1, F_2),$ *as* $N \to \infty, n \to \infty,$ *and* $m \to \infty.$

We conclude this section by presenting a corollary that plays a significant role in the two following sections.

**Corollary 3** *Under the assumption of Theorem 2,*
*i. as $a \to \infty, N \to \infty, m \to \infty,$ then,*

    *a.* $E(\mathrm{MMD}^2_{\mathrm{BNP}}(F_{1,N}^{pri}, F_{2,m})) \to 0,$ *if and only if* $H = F_2,$

    *b.* $E(\mathrm{MMD}^2_{\mathrm{BNP}}(F_{1,N}^{pos}, F_{2,m})) \to 0,$ *if and only if* $H = F_2,$

*ii. for any choice of $a$ and $H$, $E(\mathrm{MMD}^2_{\mathrm{BNP}}(F_{1,N}^{pos}, F_{2,m})) \to 0,$ if and only if $F_1 = F_2,$ as $N \to \infty,$ and $n \to \infty,$ and $m \to \infty.$*

Besides the theoretical result presented in this section, we provided the density function of the posterior-based estimator (6) compared to the baseline (3) in Figure 10 in the Appendix. This comparison, made as sample sizes approach infinity under the null hypothesis, helps to examine and understand the asymptotic distribution. The results indicated faster density convergence around zero for our proposed estimator.

## 4 Constructing a GOF Test with RB Ratio

In this section we introduce our novel semi-BNP test, utilizing the proposed estimator discussed in the previous section, to evaluate the hypothesis $\mathcal{H}_0 : F_1 = F_2$. Let the RKHS be universal and the sample space be compact, we put forward an equivalent formulation to test the hypothesis

$$\mathcal{H}_0 : \mathrm{MMD}^2(F_1, F_2) = 0, \tag{7}$$

using the RB[3] ratio, introduced by Evans (2015), as the Bayesian evidence.

By relating our problem to RB inference, with $\Psi = \mathrm{MMD}^2(F_1, F_2)$ and $\psi_0 = 0$, the RB ratio measures the change in belief regarding the true value of $\psi_0$, from *a priori* to *a posteriori*, given an observed sample $\mathbf{x}_1, \ldots, \mathbf{x}_n$ from $F_1$. It can be expressed by

$$RB_{\mathrm{MMD}^2(F_1, F_2)}(0|\mathbf{x}_{1:n}) = \frac{\pi_{\mathrm{MMD}^2(F_1, F_2)}(0|\mathbf{x}_{1:n})}{\pi_{\mathrm{MMD}^2(F_1, F_2)}(0)}, \tag{8}$$

where, $\pi_{\mathrm{MMD}^2(F_1, F_2)}(\cdot|\mathbf{x}_{1:n})$[4] and $\pi_{\mathrm{MMD}^2(F_1, F_2)}(\cdot)$ denote the probability density functions (PDFs) of the estimators given by (6) and (5), respectively.

---

[3]A detailed discussion on the RB ratio is provided in the Appendix.
[4]Note that the subscript $(F_1, F_2)$ may be omitted whenever it is clear in the context.

The density in the denominator of (8) must support $\mathcal{H}_0$ in order to reflect how well the data can support the null hypothesis based on the comparison between the prior and the posterior, utilizing the fundamental concepts of the RB ratio. Here, supporting $\mathcal{H}_0$ by $\pi_{\mathrm{MMD}^2}(\cdot)$ means to place most of the prior mass on zero. To enforce this term on $\pi_{\mathrm{MMD}^2}(\cdot)$, it is enough to set $H = F_2$ in $DP(a, H)$, which is deduced from the Theorem 1, part (iii), to make the denominator of (8) concentrate most of its mass around zero; otherwise, any other choice for $H$ contradicts the RB ratio rule in GOFs. In this case, when $\mathcal{H}_0$ is not true, for a fixed $a$ and $K$ (the upper bound of the kernel $k(\cdot, \cdot)$), the range of $\mathrm{MMD}^2_{\mathrm{BNP}}(F_{1,N}^{pri}, F_{2,m})$ should, on average, vary within a smaller range than its corresponding posterior version. Specifically, this range should be $(0, 3K)^5$, compared to $(0, \mathrm{MMD}^2(H^*, F_2) + 3K)$ which can be similarly obtained for the posterior-based MMD estimator. This indicates that $\mathcal{H}_0$ should be rejected, as it is desirable. On the other hand, when $\mathcal{H}_0$ is true, although the prior- and posterior-based MMD estimators have approximately the same range of variation $(0, 3K)$, Corollary 3(ii) implies that increasing the sample size leads the posterior to provide stronger evidence in favor of the null hypothesis compared to the prior, resulting in the acceptance of $\mathcal{H}_0$. However, the above discussion provides a general understanding rather than a tight inequality for comparison around zero. For accuracy, we use a critical value $d_{i_0/M}$ and the interval $[0, d_{i_0/M})$ to approximate the RB, addressed in (10).

With regards to choosing the concentration parameter $a$ in our proposed test, we note that $a$ controls the variation of $F^{pri}$ around $H$, which in turn controls the strength of belief in the truth of $\mathcal{H}_0$. It is recommended to choose $a < n/2$ based on the definition of $H^*$ in $F^{pos}$ (Al-Labadi & Zarepour, 2017). The idea behind using such a value of $a$ is to avoid the excessive effect of the prior $H$ on the test results by considering the chance of sampling from the observed data to be at least twice the chance of generating samples from $H$. Corollary 3(i) also clearly point to this issue in the informative prior case, as both expectations of $\mathrm{MMD}^2_{\mathrm{BNP}}(F_{1,N}^{pos}, F_{2,m})$ and $\mathrm{MMD}^2_{\mathrm{BNP}}(F_{1,N}^{pri}, F_{2,m})$ tend to 0 as $a \to \infty$, $N \to \infty$, and $m \to \infty$, iff $H = F_2$. Hence, both prior and posterior densities in (8) should be heavily massed and coincide with each other at zero. It causes the value of (8) to become very close to 1, based on which no decision can be made about $\mathcal{H}_0$.

For the proposed test, we will empirically choose $a$ to be less than $n/2$ and then compute (8). However, some computational methods in the literature have been proposed to elicit $a$ that one may be interested in using (see Al-Labadi et al., 2022b; Al-Labadi, 2021). Generally, for a given $a$, Corollary 3(ii) implies that $\mathrm{MMD}^2_{\mathrm{BNP}}(F_{1,N}^{pos}, F_{2,m})$ should be more dense than $\mathrm{MMD}^2_{\mathrm{BNP}}(F_{1,N}^{pri}, F_{2,m})$ at 0 if and only if $\mathcal{H}_0$ is true. Hence, the value of (8) presents evidence for or against $\mathcal{H}_0$, if $RB_{\mathrm{MMD}^2}(0|\mathbf{x}_{1:n}) > 1$ or $RB_{\mathrm{MMD}^2}(0|\mathbf{x}_{1:n}) < 1$, respectively. Following Evans (2015), for $t \in \mathbb{R}^+$, the calibration of (8) is defined as:

$$Str_{\mathrm{MMD}^2}(0 \,|\, \mathbf{x}_{1:n}) = \Pi_{\mathrm{MMD}^2}\big(RB_{\mathrm{MMD}^2}(t \,|\, \mathbf{x}_{1:n}) \leq RB_{\mathrm{MMD}^2}(0 \,|\, \mathbf{x}_{1:n}) \,|\, \mathbf{x}_{1:n}\big), \tag{9}$$

where, $\Pi_{\mathrm{MMD}^2}(\cdot|\mathbf{x}_{1:n})$ is the posterior probability measure corresponding to the density $\pi_{\mathrm{MMD}^2}(\cdot|\mathbf{x}_{1:n})$. When (7) is false, a small value of (9) provides strong evidence against $\psi_0$, whereas a large value suggests weak evidence against $\psi_0$. Conversely, when (7) is true, a small value of (9) indicates weak evidence in favor of $\psi_0$, while a large value suggests strong evidence in favor of $\psi_0$. Particular attention should be paid here to the computation of (8) and (9). The densities used in (8) do not have explicit forms. Thus, we use their corresponding ECDF based on $\ell$ sample sizes to estimate (8) and (9), respectively, as

$$\widehat{RB}_{\mathrm{MMD}^2}([0, \hat{d}_{i_0/M}] \,|\, \mathbf{x}_{1:n}) = \frac{\hat{\Pi}_{\mathrm{MMD}^2}(\hat{d}_{i_0/M}| \,\mathbf{x}_{1:n})}{\hat{\Pi}_{\mathrm{MMD}^2}(\hat{d}_{i_0/M})}, \tag{10}$$

$$\widehat{Str}_{\mathrm{MMD}^2}([0, \hat{d}_{i_0/M}] \,|\, \mathbf{x}_{1:n}) = \sum_D \big(\hat{\Pi}_{\mathrm{MMD}^2}(\hat{d}_{(i+1)/M}| \,\mathbf{x}_{1:n}) - \hat{\Pi}_{\mathrm{MMD}^2}(\hat{d}_{i/M}| \,\mathbf{x}_{1:n})\big), \tag{11}$$

where, $D = \left\{ i_0 \leq i \leq M - 1 : \widehat{RB}_{\mathrm{MMD}^2}\big([\hat{d}_{i/M}, \hat{d}_{(i+1)/M}] \,|\, \mathbf{x}_{1:n}\big) \leq \widehat{RB}_{\mathrm{MMD}^2}\big([0, \hat{d}_{i_0/M}] \,|\, \mathbf{x}_{1:n}\big) \right\}$, in which $M$ is a positive number, $\hat{d}_{i/M}$ is the estimate of $d_{i/M}$, the $(i/M)$-th prior quantile of (5),

$$\widehat{RB}_{\mathrm{MMD}^2}([\hat{d}_{i/M}, \hat{d}_{(i+1)/M}] \,|\, \mathbf{x}_{1:n}) = \frac{\hat{\Pi}_{\mathrm{MMD}^2}(\hat{d}_{\frac{i+1}{M}}| \,\mathbf{x}_{1:n}) - \hat{\Pi}_{\mathrm{MMD}^2}(\hat{d}_{\frac{i}{M}}| \,\mathbf{x}_{1:n})}{\hat{\Pi}_{\mathrm{MMD}^2}(\hat{d}_{\frac{i+1}{M}}) - \hat{\Pi}_{\mathrm{MMD}^2}(\hat{d}_{\frac{i}{M}})} \tag{12}$$

---

[5]This interval is the simplified form of $(0, \mathrm{MMD}^2(H, F_2) + 3K)$, where $\mathrm{MMD}^2(H, F_2)$ is zero based on setting $H = F$.

and $i_0$ in (10) is chosen so that $i_0/M$ is not too small (typically $i_0/M = 0.05$). Here $d_{i/M}$ is the threshold value that the prior-based estimator (the semi-BNP statistic under the null hypothesis) and the posterior-based estimator is compared against. Further details are available in Algorithm 1 and Figure 6 in the Appendix. For fixed $M$, as $\ell \to \infty$, then $\hat{d}_{i/M}$ converges almost surely to $d_{i/M}$ and (10) and (12) converge almost surely to $RB_{\mathrm{MMD}^2}([0, d_{i_0/M}]|\mathbf{x}_{1:n})$ and $RB_{\mathrm{MMD}^2}([d_{i/M}, d_{(i+1)/M}]|\mathbf{x}_{1:n})$, respectively. These are obtained from the left Riemann sum approximation and the relationship between CDFs and PDFs with considering $d_{i/M} + \epsilon = d_{(i+1)/M}$ for any small $\epsilon > 0$, as follows:

$$RB_{\mathrm{MMD}^2}([d_{\frac{i}{M}}, d_{\frac{i+1}{M}}]|\mathbf{x}_{1:n}) = \frac{\int_{d_{\frac{i}{M}}}^{d_{\frac{i+1}{M}}} \pi_{\mathrm{MMD}^2}(t|\mathbf{x}_{1:n})\,dt}{\int_{d_{\frac{i}{M}}}^{d_{\frac{i+1}{M}}} \pi_{\mathrm{MMD}^2}(t)\,dt} = \frac{\epsilon \pi_{\mathrm{MMD}^2}(d_{\frac{i}{M}}|\mathbf{x}_{1:n})}{\epsilon \pi_{\mathrm{MMD}^2}(d_{\frac{i}{M}})} \approx RB_{\mathrm{MMD}^2}(d_{i/M}|\mathbf{x}_{1:n}).$$

Consequently, (11) converges almost surly to $Str_{\mathrm{MMD}^2}([0, d_{i_0/M}]\,|\,\mathbf{x}_{1:n}) = \sum_{D'} \int_{d_{i/M}}^{d_{(i+1)/M}} \pi_{\mathrm{MMD}^2}(t|\mathbf{x}_{1:n})\,dt$, where $D'$ is defined similarly to $D$ by replacing $\widehat{RB}$ and $\hat{d}_{i/M}$ with $RB$ and $d_{i/M}$ in $D$. The following result from Al-Labadi & Evans (2018, Proposition 6) gives the consistency of the proposed test. If $\mathcal{H}_0$ is true, then $RB_{\mathrm{MMD}^2}([0, d_{i_0/M}]|\mathbf{x}_{1:n}) \xrightarrow{a.s.} M/i_0(> 1)$ and $RB_{\mathrm{MMD}^2}([d_{i/M}, d_{(i+1)/M}]|\mathbf{x}_{1:n}) \xrightarrow{a.s.} 0$ for $i_0 \leq i \leq M - 1$, as $n \to \infty$, which implies that $Str_{\mathrm{MMD}^2}([0, d_{i_0/M}]\,|\,\mathbf{x}_{1:n})$ converges to 1 almost surely. Otherwise, both $RB_{\mathrm{MMD}^2}([0, d_{i_0/M}]\,|\,\mathbf{x}_{1:n})$ and $Str_{\mathrm{MMD}^2}([0, d_{i_0/M}]\,|\,\mathbf{x}_{1:n})$ converge to 0.

The proposed test is suggested to overcome several limitations present in its frequentist counterparts. In a frequentist test, for a given permissible type I error rate denoted by $\alpha$, the test rejects $\mathcal{H}_0$ if the value of $\mathrm{MMD}^2(F_1, F_2)$ is greater than some threshold $c_\alpha$. The corresponding $p$-value for this test can also be computed by $\Pr(\mathrm{MMD}^2(F_1, F_2) \geq c_\alpha | \mathcal{H}_0)$, which leads the test to reject $\mathcal{H}_0$ if it is less than $\alpha$. However, Li et al. (2017) noted that if $\mathrm{MMD}^2(F_1, F_2)$ is not significantly larger than $c_\alpha$ for some finite samples when $\mathcal{H}_0$ is not true, the null hypothesis $\mathcal{H}_0$ is not rejected. Furthermore, there is a trade-off between the permissible type I error rate $\alpha$ and the probability of failing to reject a false null hypothesis (type II error), denoted by $\beta$, as $\alpha + \beta \leq 1$. Decreasing one error rate inevitably leads to an increase in the other, indicating that we cannot arbitrarily drive to type I error rate to zero. Moreover, the $p$-values are uniformly distributed between 0 and 1 under the null hypothesis. In fact, it does not allow evidence for the null, which is one of their weaknesses compared to Bayesian criteria in hypothesis testing problems.

# 5 Embedding the Semi-BNP Estimator in GAN Learning

In this section, we propose a BNPL procedure that leverages a posterior-based MMD estimator to train GANs. It is inspired by the idea presented in Dellaporta et al. (2022) to approximate the posterior on the generator's parameters.

## 5.1 Generative Adversarial Networks

The GAN (Goodfellow et al., 2014) is a machine learning technique used to generate realistic-looking artificial samples. In this context, the discriminator $D$ can be viewed as a black box that uses a discrepancy measure $\delta$ to differentiate between the real and fake data. Meanwhile, the generator $G_{\boldsymbol{\omega}}$ is trained by optimizing a simpler objective function, given by

$$\arg \min_{\boldsymbol{\omega} \in \mathcal{W}} \delta(F, F_{G_{\boldsymbol{\omega}}}),$$

where $F_{G_{\boldsymbol{\omega}}}$ represents the distribution of the generator. In fact, $D$ attempts to continuously train $G_{\boldsymbol{\omega}}$ by computing distance $\delta$ between $F$ and $F_{G_{\boldsymbol{\omega}}}$ until this distance is negligible, making their difference indistinguishable. This technique leads to omitting the neural network from $D$, whose optimization may lead to a vanishing gradient. An effective measure of discrepancy for $\delta$ is the MMD, which is a kernel-based measure that offers several desirable properties such as consistency and robustness in generating samples (Gretton et al., 2012a; Chérief-Abdellatif & Alquier, 2022).

Numerous frequentist GANs applying the MMD measure to estimate the generator's parameters can be found in the literature. (Dziugaite et al., 2015; Bińkowski et al., 2018; Li et al., 2015). These models are devised by comparing the generated fake samples with real samples. In addition to the MMD, several other discrepancy measures are commonly used for GANs, including the $f$-divergence measure (Nowozin et al., 2016), the Wasserstein distance (Arjovsky et al., 2017), and the total variation distance (Lin et al., 2018). Nevertheless, the MMD kernel-based measure is remarkably robust against outliers and has the exceptional ability to capture complex dependencies in the data (Sejdinovic et al., 2013; Chérief-Abdellatif & Alquier, 2022). This makes it highly effective in handling model misspecification and detecting subtle differences between distributions. This property is particularly useful for modeling complicated datasets such as images, which are a common application for GANs. Moreover, Al-Labadi et al. (2022a) used the energy distance to expand their procedure, which is a member of the larger class of MMD kernel-based measures (Sejdinovic et al., 2013). From here, it is obvious that choosing among a larger class can lead to designing more sensitive discrepancy measures to detect differences.

Although a particular case of the test in Al-Labadi et al. (2022a) can be used to compare two distributions, it cannot be easily used as a discriminator in the minimum distance estimation technique to train GANs. In GANs, the objective is to update the parameter $\boldsymbol{\omega}$ of the deterministic generative neural network $G_{\boldsymbol{\omega}}$. Therefore, treating $F_{G_{\boldsymbol{\omega}}}$ as an unknown distribution on which we place a BNP prior is nonsensical. Consequently, a more suitable distance criterion is required to compare an intractable parametric distribution with an unknown distribution.

## 5.2 Architecture

Various GAN architectures can be found in the literature to model complex high-dimensional distributions. However, we consider the original architecture of the generator network in Goodfellow et al. (2014). Specifically, we follow the neural network architecture in Goodfellow et al. by setting the generator, $G_{\boldsymbol{\omega}}$, to be a multi-layer neural network with parameters $\boldsymbol{\omega}$, rectified linear units activation function for each hidden layer, and a sigmoid function for the last layer (output layer). The generator receives a noise vector $\boldsymbol{U} = (U_1, \ldots, U_p)$ as its input nodes, where $p < d$, and each element of $\boldsymbol{U}$ is independently drawn from the same distribution $F_U$.

Our discriminator follows the work of Dziugaite et al. (2015); Li et al. (2015) by optimizing the objective function:

$$\arg \min_{\boldsymbol{\omega} \in \mathcal{W}} \mathrm{MMD}^2_{\mathrm{BNP}}(F_N^{pos}, F_{G_{\boldsymbol{\omega}}, m}),$$

but we now estimate the MMD using our semi-BNP method. In fact, our BNPL procedure implicitly approximates samples from the posterior distribution on the parameter $\boldsymbol{\omega}$ by minimizing the posterior-based MMD estimator. For any differentiable kernel function $k(\cdot, \cdot)$, this optimization is performed by computing the following gradient based on samples from $F|\mathbf{X}_{1:n} \sim DP(a + n, H^*)$, as

$$\frac{\partial \mathrm{MMD}^2_{\mathrm{BNP}}(F_N^{pos}, F_{G_{\boldsymbol{\omega}}, m})}{\partial \boldsymbol{\omega}_i} = \sum_{\ell=1}^{N} \sum_{t=1}^{m} \left\{ \frac{\partial}{\partial \mathbf{Y}_t} \left[ -\frac{2}{m} \sum_{t=1}^{m} J_{\ell,N}^* k(\mathbf{V}_\ell^*, \mathbf{Y}_t) \right.\right.$$
$$\left.\left. + \frac{1}{Nm^2} \sum_{t,t'=1}^{m} k(\mathbf{Y}_t, \mathbf{Y}_{t'}) \right] \frac{\partial \mathbf{Y}_t}{\partial \boldsymbol{\omega}} \right\},$$

where, $\mathbf{Y}_t = G_{\boldsymbol{\omega}}(\boldsymbol{U}_t)$, $\boldsymbol{U}_t = (U_{t1}, \ldots, U_{tp})$, and $U_{ti}$'s are generated from a distribution $F_U$, for $t = 1, \ldots, m$, and $i = 1, \ldots, p$. Then, the backpropagation method is applied for calculating partial derivatives $\frac{\partial \mathbf{Y}_t}{\partial \boldsymbol{\omega}}$ to update the parameters of $G_{\boldsymbol{\omega}}$.

However, Li et al. (2015, Equation 8) remarked that considering the square root of the MMD measure given by (2) in the cost function of frequentist GANs is more efficient than using (2) to train network $G_{\boldsymbol{\omega}}$. They mentioned that since the gradient of $\sqrt{\mathrm{MMD}^2(F_N, F_{G_{\boldsymbol{\omega}}, m})}$ with respect to $\boldsymbol{\omega}$ is the product of $\gamma_1 = \frac{1}{2\sqrt{\mathrm{MMD}^2(F_N, F_{G_{\boldsymbol{\omega}}, m})}}$ and $\gamma_2 = \frac{\partial \mathrm{MMD}^2(F_N, F_{G_{\boldsymbol{\omega}}, m})}{\partial \boldsymbol{\omega}}$, then $\gamma_1$ forces the value of the gradient to be relatively

large, even if both $\mathrm{MMD}^2(F_N, F_{G_{\boldsymbol{\omega}},m})$ and $\gamma_2$ are small. This can prevent the vanishing gradient, which improves the learning of the parameters of $G_{\boldsymbol{\omega}}$ in the early layers of this network. We consider this point in order to improve our semi-BNP objective function:

$$\arg \min_{\boldsymbol{\omega} \in \mathcal{W}} \mathrm{MMD}_{\mathrm{BNP}}(F_N^{pos}, F_{G_{\boldsymbol{\omega}},m}). \tag{13}$$

Algorithm 2 in the Appendix provides steps for implementing the training.

Let $\boldsymbol{\omega}^*$ be the optimized parameter of $G_{\boldsymbol{\omega}}$ that minimizes $\mathrm{MMD}_{\mathrm{BNP}}(F_N^{pos}, F_{G_{\boldsymbol{\omega}},m})$. Since $\mathrm{MMD}_{\mathrm{BNP}}(F_N^{pos}, F_{G_{\boldsymbol{\omega}},m})$ can be viewed as a semi-BNP estimation of (2), it becomes imperative to assess the accuracy of this estimation, specifically in terms of how effectively the proposed GAN can generate realistic samples that faithfully represent the true data distribution (generalization error). Furthermore, it is crucial to take into consideration the generator's performance in dealing with outliers which includes a small proportion of observations that deviate from the clean data distribution $F_0$ (robustness). The next lemma addresses these two concerns.

**Lemma 4** *Let $\mathcal{W}$ be the parameter space for $G_{\boldsymbol{\omega}}$ and $\boldsymbol{\omega}^* \in \mathcal{W}$ be the value that optimizes the objective function (13) and $\boldsymbol{\omega}'$ be the true value that minimizes $\mathrm{MMD}(F, F_{G_{\boldsymbol{\omega}}})$. Assume that $F \sim DP(a, H)$ and let $k(\cdot, \cdot)$ be any continuous kernel function with feature space corresponding to a universal RKHS defined on a compact metric space $\mathfrak{X}$ such that $|k(\boldsymbol{z}, \boldsymbol{z}')| < K$, for any $\boldsymbol{z}, \boldsymbol{z}' \in \mathbb{R}^d$. For a given sample $\mathbf{X}_1, \ldots, \mathbf{X}_n$ from distribution $F$:*
*i. Generalization error:*

$$E\left(\mathrm{MMD}(F, F_{G_{\boldsymbol{\omega}^*}})\right) \leq \mathrm{MMD}(F, F_{G_{\boldsymbol{\omega}'}}) + \frac{2K}{\sqrt{n}} + \frac{4aK}{a+n} + 2\sqrt{\frac{(a+n+N)K}{(a+n+1)N}}.$$

*ii. Robustness: Suppose there exist outliers in the sample data, which arise from a noise distribution $Q$. Consider the Hüber's contamination model (Huber, 1992; Chérief-Abdellatif & Alquier, 2022), given by $F = (1-\epsilon)F_0 + \epsilon Q$, where $\epsilon \in (0, \frac{1}{2})$ is the contamination rate, and the latent variables $Z_1, \ldots, Z_n \overset{i.i.d.}{\sim} \mathrm{Bernoulli}(\epsilon)$ are such that $\mathbf{X}_i \overset{i.i.d.}{\sim} F_0$ if $Z_i = 0$; otherwise, $\mathbf{X}_i \overset{i.i.d.}{\sim} Q$. Then,*

$$E\left(\mathrm{MMD}(F_0, F_{G_{\boldsymbol{\omega}^*}})\right) \leq \min_{\boldsymbol{\omega} \in \mathcal{W}} \mathrm{MMD}(F_0, F_{G_{\boldsymbol{\omega}}}) + 4\epsilon + \frac{2K}{\sqrt{n}} + \frac{4aK}{a+n} + 2\sqrt{\frac{(a+n+N)K}{(a+n+1)N}}.$$

Lemma 4(ii) demonstrates that despite encountering outlier data, $F_{G_{\boldsymbol{\omega}^*}}$ and $F_0$ are negligibly different for a sufficiently large sample size. This feature results in the majority of the posterior on the parameter space $\mathcal{W}$ being distributed on value $\boldsymbol{\omega}^*$, which is a desirable outcome of the proposed method.

Although the preceding statements investigate properties of the estimated parameters by providing upper bounds for the expectation of the MMD estimator, the next lemma presents stochastic bounds for the estimation error in order to assess the posterior consistency.

**Lemma 5** *Building upon the general assumptions stated in Lemma 4, for a given sample $\mathbf{X}_1, \ldots, \mathbf{X}_n$ from distribution $F$ in the probability space $(\mathfrak{X}, \mathcal{A}, \mathrm{Pr})$ and any $\epsilon > 0$,*
*i. $\mathrm{Pr}\left(|\mathrm{MMD}(F_N^{pos}, F_{G_{\boldsymbol{\omega}^*},m}) - \mathrm{MMD}(F, F_{G_{\boldsymbol{\omega}'}})| \geq h(n, m, K, \epsilon) + |\Delta_1| + |\Delta_2|\right) \leq 2\exp\frac{-\epsilon^2 nm}{2K(n+m)}$,*

*ii. $\mathrm{Pr}\left(\mathrm{MMD}(F, F_{G_{\boldsymbol{\omega}^*}}) > \epsilon\right) \leq \frac{1}{\epsilon}\left(\mathrm{MMD}(F, F_{G_{\boldsymbol{\omega}'}}) + \frac{2K}{\sqrt{n}} + \frac{4aK}{a+n} + 2\sqrt{\frac{(a+n+N)K}{(a+n+1)N}}.\right)$,*

*where, $h(n, m, K, \epsilon) = 2\sqrt{K}(\sqrt{n} + \sqrt{m})/\sqrt{nm} + \epsilon$, $\Delta_1 = \mathrm{MMD}(F_N^{pos}, F_{G_{\boldsymbol{\omega}^*}}) - \mathrm{MMD}(F_n, F_{G_{\boldsymbol{\omega}'},m})$, and $\Delta_2 = \mathrm{MMD}(F, F_{G_{\boldsymbol{\omega}^*}}) - \mathrm{MMD}(F, F_{G_{\boldsymbol{\omega}'}})$.*

A direct consequence of Lemma 5(ii) is that for a fixed value of $a$, $\mathrm{Pr}(\mathrm{MMD}(F, F_{G_{\boldsymbol{\omega}^*}}) \geq \epsilon) \to 0$, as $n \to \infty$ and $N \to \infty$, for any $\epsilon > 0$, when $\mathrm{MMD}(F, F_{G_{\boldsymbol{\omega}'}}) = 0$ (well-specified case). This implies $F_{G_{\boldsymbol{\omega}^*}}$ converges in probability to the data distribution $F$ as the sample size increases in well-specified cases.

The choice of $a$ in the test proposed in Section 4 plays a crucial role in determining the degree of support for the null hypothesis against the alternative. In the context of approximating the posterior on the parameter space, the prior choice for $F$ and determining the strength of belief becomes challenging. We consider a small value for $a$ as a non-informative prior, following the suggestion by Dellaporta et al. (2022), thanks to its broad ability to characterize uncertainty (Terenin & Draper, 2017). However, it's important to note that setting $a = 0$ as done by Dellaporta et al. (2022) is not always well-defined mathematically, as the DP is only defined for $a > 0$. Therefore, we opt for $a = 10^{-6}$.

The main distinction between our BNPL method and the one proposed by Dellaporta et al. (2022) lies in the fact that we generalize their BNPL procedure beyond estimating parameters and explicitly consider the terms of the DP posterior approximation and their corresponding weights. Dellaporta et al. used the following DP approximation:

$$F_{n+N}^{pos} = \sum_{\ell=1}^{n} \widetilde{J}_{\ell,n} \delta_{\mathbf{X}_\ell} + \sum_{t=1}^{N} J_{t,N} \delta_{\mathbf{V}_t},$$

where $(\widetilde{J}_{1:n,n}, J_{1:N,N}) \sim \text{Dirichlet}(1, \ldots, 1, \frac{a}{N}, \ldots, \frac{a}{N})$, $(\mathbf{X}_{1:n}) \overset{i.i.d.}{\sim} F$, and $(\mathbf{V}_{1:N}) \overset{i.i.d.}{\sim} H$. In contrast, we employ $F_N^{pos} = \sum_{i=1}^{N} J_{i,N}^* \delta_{V_i^*}$, with $(J_{1:N,N}^*) \sim \text{Dirichlet}(\frac{a+n}{N}, \ldots, \frac{a+n}{N})$. Our approach offers an advantage over the approximation used in Dellaporta et al. (2022) due to its reduced number of terms, significantly reducing both computational and theoretical complexity, where our method scales $O(N)$ and Dellaporta et al. scales $O(N + n)$. Additionally, a further difference is that Dellaporta's bootstrap procedure needs to query the loss function $B$ times to simulate $B$ posterior parameters, whereas our procedure does not require a bootstrap algorithm and we only need to simulate a single parameter. Although their bootstrap procedure is embarrassingly parallelizable, the number of bootstrap samples generally should be a fairly large number and the typical statistical practitioner does not have access to $B$ cores to truly parallelize the additional cost of bootstrap sampling.

### 5.3 Kernel Settings

In our method, we choose to use the standard radial basis function (RBF) kernel as its feature space corresponds to a universal RKHS. For a comprehensive understanding of RBF functions, refer to Section D in the Appendix. Dziugaite et al. (2015); Li et al. (2015) and Li et al. (2017) used the Gaussian kernel in training MMD-GANs because of its simplicity and good performance. Dziugaite et al. (2015) also evaluated some other RBF kernels such as the Laplacian and rational quadratic kernels to compare the results of the MMD-GANs with those obtained based on using Gaussian kernels. They found the best performance by applying the Gaussian kernel in the MMD cost function.

Hence, we consider the Gaussian kernel function in our proposed procedure. To choose the bandwidth parameter $\sigma$, we follow the idea of considering a set of fixed values of $\sigma$'s such as $\{\sigma_1, \ldots, \sigma_T\}$, then compute the mixture of Gaussian kernels $k(\cdot, \cdot) = \sum_{t=1}^{T} k_{G_{\sigma_t}}(\cdot, \cdot)$, to consider in (6). For each $\sigma(t)$, $0 \leq k_{G_{\sigma_t}}(\cdot, \cdot) \leq 1$; hence, $0 \leq k(\cdot, \cdot) \leq T$, which satisfies the theoretical results presented in the paper. As it is mentioned in Li et al. (2015), this choice reflects a good performance in training MMD-GANs.

## 6 Experimental Investigation

In this section, we empirically investigate our proposed methods through comprehensive numerical studies in the following two subsections, which demonstrate the superior performance of our proposed semi-BNP test as a standalone test as well as an embedded discriminator for the semi-BNP GAN [6].

### 6.1 The Semi-BNP Test

To comprehensively study test performance evaluation, we consider some major representative examples in two-sample comparison problems. For this, let $\mathbf{y}_1, \ldots, \mathbf{y}_n$ be a sample generated from $F_2 = N(\mathbf{0}_d, I_d)$ and

---

[6]The relevant codes for the semi-BNP procedure are available at `https://github.com/ForoughFazeliasl/SemiBNPMMD.git`.

$\mathbf{x}_1, \ldots, \mathbf{x}_n$ be an observed sample generated from each below distributions: $F_1 = N(\mathbf{0}_d, I_d)$ (No differences), $F_1 = N(\mathbf{0.5}_d, I_d)$ (Mean shift), $F_1 = LN(\mathbf{0}_d, B_d)$ (Skewness), $F_1 = \frac{1}{2}N(-\mathbf{1}_d, I_d) + \frac{1}{2}N(\mathbf{1}_d, I_d)$ (Mixture), $F_1 = N(\mathbf{0}_d, 2I_d)$ (Variance shift), $F_1 = t_3(\mathbf{0}_d, I_d)$ (Heavy tail), and $F_1 = LG(\mathbf{0}_d, I_d)$ (Kurtosis).

To implement the test, we set $\ell = 1000$, $M = 20$, and $\epsilon = 10^{-3}$ to be used in Algorithm 1 in the Appendix. We first considered the mixture of six Gaussian kernels corresponding to the suggested bandwidth parameters $2, 5, 10, 20, 40$, and $80$ by Li et al. (2015). We found that although this choice can provide good results in training GANs, it does not provide satisfactory results in hypothesis testing problems.

Instead of using a mixture of several Gaussian kernels, we propose choosing a specific value for the bandwidth parameter that maximizes the area under the receiver operating characteristic curve (AUC) empirically.

In a binary classifier, which can also be thought of as a two-sample test assessing whether two samples are distinguishable or not, the receiver operating characteristic (ROC) curve is a plot of true positive rates (sensitivity) against the false positive rates (1-specificity) based on different choices of threshold to display the performance of the test. The positive term refers to rejecting $\mathcal{H}_0$ in (7), while, the negative term refers to failing to reject $\mathcal{H}_0$. The false positive and false negative rates are equivalent to type I and type II errors, respectively. Hence, a higher AUC indicates a better diagnostic ability of a binary test[7].

The ROC curves and AUC values of the synthetic examples are provided in Figure 1 for the sample size $n = 50$, $d = 60$, $a = 25$, and various values of the bandwidth parameter, including the median heuristic $\sigma_{MH}$. The red diagonal line represents the random classifier. An ROC curve located higher than the diagonal line indicates better test performance and vice versa. It is obvious from Figure 1 that the best test performance (AUC $= 1$) is first achieved for the bandwidth parameter $80$. For this $\sigma$, the AUC jumps from zero to 1 as either $\mathcal{H}_0$ is strongly rejected by too small values of RB (near to 0, the minimum RB) in alternative experiments or $\mathcal{H}_0$ is strongly accepted by large values of RB (near to 20, the maximum RB) in null experiments. As noted in the Appendix, if $\sigma$ is too small or too large, the MMD approaches zero, resulting in poor performance, as shown in Figure 1. In the context of this paper's semi-BNP test, both the prior- and posterior-based MMD estimators converge to zero under these conditions, causing their corresponding density functions to coincide at zero, resulting in $RB \approx 1$ and rendering the test unable to evaluate $\mathcal{H}_0$.

Another test of interest is to assess the effect of different hyperparameter settings for $a$ and $H$ through simulation studies to follow our proposed theoretical convergence results. To do this, we generate 100 60-dimensional samples of sizes $n = 50$ from both $F_1 = t_3(\mathbf{0}_{60}, I_{60})$ and $F_2 = N(\mathbf{0}_{60}, I_{60})$ and represent the result of the semi-BNP test by Figure 2 for two choices of the base measure $H$ ($H = F_2$ and $H = LG(\mathbf{0}_{60}, I_{60})$) and various values of $a$ ($a = 1, \ldots, 1000$). In this figure, the solid line represents the average of the RB and the filled area around the line indicates a 95% confidence interval of the RB over the 100 samples. Figure 2-a clearly shows that by choosing $H \neq F_2$, the test wrongly accepts the null hypothesis. It is because the prior does not support the null hypothesis mentioned earlier when presenting the RB ratio in Section 4. On the other hand, when $H = F_2$, Figure 2-b shows good performance for the test at $a = n/2$. Failing to reject $\mathcal{H}_0$ for small values of $a$ is due to the lack of sufficient support from the null hypothesis by the prior. We remark that the value of $a$ determines the concentration of the prior $F^{pri}$ around $H$, thus it is obvious that for small values of $a$, the test does not perform well. It should also be noted that for any choices of $H$ in Figure 2, the ability of the test to evaluate the null hypothesis is reduced by letting $a$ go to infinity, which can be concluded by Corollary 3(i).

Now, to conduct a more comprehensive investigation, we present the average of RB and its relevant strength over the 100 samples in Table 1 for $n = 30, 50$. Furthermore, we present the results of the BNP-energy test by Al-Labadi et al. (2022a) in Table 1, which demonstrate its weak performance in certain scenarios. Additional results in the power comparison can be found in Section F.1 of the Appendix.

---

[7]Since we consider $i_0/M = 0.05$ to estimate the RB ratio, the values of $RB$ can vary between 0 and 20. Therefore, in computing the AUC for the semi-BNP test, the threshold should vary from 0 to 20. More details for plotting the ROC and computing the AUC are provided by Algorithm 3 in the Appendix.

Table 1: The average of RB, the average of its strength (Str ), and the relevant AUC out of 100 replications based on using $a = 25$, $\ell = 1000$, $M = 20$, $\epsilon = 10^{-3}$ in (4), and bandwidth parameter $\sigma = 80$ in RBF kernel for two sample of data with $n = 30, 50$.

| Example | $d$ | BNP MMD RB(Str) 30 | BNP MMD RB(Str) 50 | BNP MMD AUC 30 | BNP MMD AUC 50 | BNP Energy RB(Str) 30 | BNP Energy RB(Str) 50 | BNP Energy AUC 30 | BNP Energy AUC 50 | FNP MMD P.value 30 | FNP MMD P.value 50 | FNP MMD AUC 30 | FNP MMD AUC 50 | FNP Energy P.value 30 | FNP Energy P.value 50 | FNP Energy AUC 30 | FNP Energy AUC 50 |
|---|---|---|---|---|---|---|---|---|---|---|---|---|---|---|---|---|---|
| No differences | 1 | 2.08(0.62) | 2.41(0.67) | | | 1.78(0.59) | 1.91(0.55) | | | 0.50 | 0.45 | | | 0.50 | 0.49 | | |
| | 5 | 4.06(0.77) | 6.91(0.76) | | | 3.46(0.65) | 5.99(0.73) | | | 0.48 | 0.50 | | | 0.54 | 0.52 | | |
| | 10 | 6.21(0.78) | 10.74(0.79) | | | 5.92(0.67) | 10.42(0.76) | | | 0.50 | 0.51 | | | 0.54 | 0.47 | | |
| | 20 | 9.62(0.80) | 16.02(0.83) | | | 8.24(0.73) | 14.76(0.78) | | | 0.46 | 0.50 | | | 0.51 | 0.50 | | |
| | 40 | 13.07(0.88) | 18.85(0.97) | | | 11.56(0.75) | 17.58(0.84) | | | 0.51 | 0.49 | | | 0.53 | 0.46 | | |
| | 60 | 14.09(0.87) | 19.71(1) | | | 13.38(0.81) | 18.51(0.93) | | | 0.52 | 0.46 | | | 0.50 | 0.48 | | |
| | 80 | 15.2(0.89) | 19.57(1) | | | 14.16(0.87) | 19.10(1) | | | 0.46 | 0.47 | | | 0.53 | 0.56 | | |
| | 100 | 15.83(0.91) | 19.74(1) | | | 14.84(0.92) | 19.31(1) | | | 0.48 | 0.46 | | | 0.49 | 0.55 | | |
| Mean shift | 1 | 0.76(0.24) | 0.40(0.09) | 0.82 | 0.96 | 0.67(0.21) | 0.45(0.11) | 0.87 | 0.90 | 0.15 | 0.05 | 0.86 | 0.91 | 0.19 | 0.12 | 0.79 | 0.86 |
| | 5 | 0.21(0.03) | 0.07(0) | 0.99 | 0.99 | 0.28(0.04) | 0.09(0.01) | 0.98 | 1 | 0.01 | 0.002 | 1 | 0.98 | 0.02 | 0.004 | 0.97 | 0.97 |
| | 10 | 0.09(0.01) | 0.05(0) | 1 | 1 | 0.17(0.05) | 0.02(0) | 0.98 | 1 | 0.001 | 0.001 | 1 | 1 | 0.006 | 0.004 | 0.98 | 1 |
| | 20 | 0.09(0.01) | 0(0) | 1 | 1 | 0.09(0.01) | 0(0) | 1 | 1 | 0.001 | 0.001 | 1 | 1 | 0.004 | 0.004 | 1 | 1 |
| | 40 | 0.08(0) | 0(0) | 1 | 1 | 0.06(0.02) | 0(0) | 1 | 1 | 0.001 | 0.001 | 1 | 1 | 0.004 | 0.004 | 1 | 1 |
| | 60 | 0.09(0.03) | 0(0) | 1 | 1 | 0.07(0.04) | 0(0) | 1 | 1 | 0.001 | 0.001 | 1 | 1 | 0.004 | 0.004 | 1 | 1 |
| | 80 | 0.06(0.02) | 0(0) | 1 | 1 | 0.05(0.03) | 0(0) | 1 | 1 | 0.001 | 0.001 | 1 | 1 | 0.004 | 0.004 | 1 | 1 |
| | 100 | 0.04(0.01) | 0(0) | 1 | 1 | 0.03(0) | 0(0) | 1 | 1 | 0.001 | 0.001 | 1 | 1 | 0.004 | 0.004 | 1 | 1 |
| Skewness | 1 | 0.01(0) | 0(0) | 0.99 | 1 | 0.07(0) | 0(0) | 0.99 | 1 | 0.009 | 0.001 | 0.98 | 1 | 0.007 | 0.004 | 0.94 | 1 |
| | 5 | 0(0) | 0(0) | 1 | 1 | 0(0) | 0(0) | 1 | 1 | 0.001 | 0.001 | 1 | 1 | 0.004 | 0.004 | 1 | 1 |
| | 10 | 0(0) | 0(0) | 1 | 1 | 0(0) | 0(0) | 1 | 1 | 0.001 | 0.001 | 1 | 1 | 0.004 | 0.004 | 1 | 1 |
| | 20 | 0(0) | 0(0) | 1 | 1 | 0(0) | 0(0) | 1 | 1 | 0.001 | 0.001 | 1 | 1 | 0.004 | 0.004 | 1 | 1 |
| | 40 | 0(0) | 0(0) | 1 | 1 | 0(0) | 0(0) | 1 | 1 | 0.001 | 0.001 | 1 | 1 | 0.004 | 0.004 | 1 | 1 |
| | 60 | 0(0) | 0(0) | 1 | 1 | 0(0) | 0(0) | 1 | 1 | 0.001 | 0.001 | 1 | 1 | 0.004 | 0.004 | 1 | 1 |
| | 80 | 0(0) | 0(0) | 1 | 1 | 0(0) | 0(0) | 1 | 1 | 0.001 | 0.001 | 1 | 1 | 0.004 | 0.004 | 1 | 1 |
| | 100 | 0(0) | 0(0) | 1 | 1 | 0(0) | 0(0) | 1 | 1 | 0.001 | 0.001 | 1 | 1 | 0.004 | 0.004 | 1 | 1 |
| Mixture | 1 | 0.06(0) | 0(0) | 0.90 | 0.97 | 0.19(0.03) | 0.04(0) | 0.97 | 1 | 0.43 | 0.38 | 0.58 | 0.57 | 0.29 | 0.17 | 0.69 | 0.81 |
| | 5 | 0(0) | 0(0) | 1 | 1 | 0(0) | 0(0) | 1 | 1 | 0.15 | 0.09 | 0.84 | 0.91 | 0.06 | 0.01 | 0.95 | 1 |
| | 10 | 0(0) | 0(0) | 1 | 1 | 0(0) | 0(0) | 1 | 1 | 0.03 | 0.007 | 0.95 | 0.98 | 0.02 | 0.007 | 0.96 | 1 |
| | 20 | 0(0) | 0(0) | 1 | 1 | 0(0) | 0(0) | 1 | 1 | 0.002 | 0.001 | 0.96 | 1 | 0.01 | 0.006 | 1 | 1 |
| | 40 | 0(0) | 0(0) | 1 | 1 | 0(0) | 0(0) | 1 | 1 | 0.001 | 0.001 | 1 | 1 | 0.01 | 0.006 | 1 | 1 |
| | 60 | 0(0) | 0(0) | 1 | 1 | 0(0) | 0(0) | 1 | 1 | 0.001 | 0.001 | 1 | 1 | 0.006 | 0.009 | 1 | 1 |
| | 80 | 0(0) | 0(0) | 1 | 1 | 0(0) | 0(0) | 1 | 1 | 0.001 | 0.001 | 1 | 1 | 0.008 | 0.006 | 1 | 1 |
| | 100 | 0(0) | 0(0) | 1 | 1 | 0(0) | 0(0) | 1 | 1 | 0.001 | 0.001 | 1 | 1 | 0.004 | 0.006 | 1 | 1 |
| Variance shift | 1 | 0.87(0.29) | 0.85(0.19) | 0.71 | 0.83 | 1.10(0.36) | 1.08(0.33) | 0.53 | 0.63 | 0.46 | 0.38 | 0.54 | 0.57 | 0.33 | 0.21 | 0.65 | 0.77 |
| | 5 | 0.55(0.12) | 0.56(0.15) | 0.99 | 0.99 | 1.06(0.35) | 0.99(0.32) | 0.89 | 0.98 | 0.34 | 0.20 | 0.65 | 0.80 | 0.20 | 0.07 | 0.82 | 0.93 |
| | 10 | 0.44(0.11) | 0.27(0.05) | 0.99 | 1 | 0.87(0.24) | 0.80(0.25) | 0.97 | 1 | 0.14 | 0.03 | 0.85 | 0.97 | 0.10 | 0.02 | 0.89 | 0.97 |
| | 20 | 0.34(0.07) | 0.08(0) | 1 | 1 | 0.65(0.17) | 0.60(0.13) | 0.99 | 1 | 0.01 | 0.001 | 0.95 | 1 | 0.03 | 0.006 | 0.95 | 1 |
| | 40 | 0.13(0.01) | 0.02(0) | 1 | 1 | 0.61(0.18) | 0.58(0.14) | 1 | 1 | 0.001 | 0.001 | 1 | 1 | 0.01 | 0.004 | 0.98 | 1 |
| | 60 | 0.12(0.01) | 0.01(0) | 1 | 1 | 0.47(0.10) | 0.45(0.11) | 1 | 1 | 0.001 | 0.001 | 1 | 1 | 0.006 | 0.004 | 1 | 1 |
| | 80 | 0.17(0.01) | 0(0) | 1 | 1 | 0.54(0.12) | 0.47(0.11) | 1 | 1 | 0.001 | 0.001 | 1 | 1 | 0.005 | 0.004 | 1 | 1 |
| | 100 | 0.14(0.01) | 0(0) | 1 | 1 | 0.45(0.10) | 0.41(0.08) | 1 | 1 | 0.001 | 0.001 | 1 | 1 | 0.004 | 0.004 | 1 | 1 |
| Heavy tail | 1 | 0.93(0.28) | 0.66(0.20) | 0.89 | 0.92 | 1.19(0.41) | 1.10(0.38) | 0.70 | 0.78 | 0.43 | 0.39 | 0.57 | 0.56 | 0.39 | 0.36 | 0.59 | 0.62 |
| | 5 | 0.32(0.06) | 0.37(0.08) | 0.99 | 0.99 | 0.77(0.24) | 0.78(0.23) | 0.93 | 0.99 | 0.20 | 0.11 | 0.79 | 0.89 | 0.03 | 0.006 | 0.97 | 0.99 |
| | 10 | 0.35(0.08) | 0.13(0.02) | 0.99 | 1 | 0.61(0.16) | 0.68(0.19) | 0.98 | 1 | 0.06 | 0.007 | 0.92 | 0.98 | 0.09 | 0.01 | 0.90 | 0.97 |
| | 20 | 0.15(0.02) | 0(0) | 1 | 1 | 0.48(0.12) | 0.46(0.12) | 1 | 1 | 0.002 | 0.001 | 0.96 | 1 | 0.02 | 0.005 | 0.96 | 1 |
| | 40 | 0.07(0.01) | 0(0) | 1 | 1 | 0.25(0.04) | 0.18(0.04) | 1 | 1 | 0.001 | 0.001 | 1 | 1 | 0.005 | 0.004 | 1 | 1 |
| | 60 | 0.02(0) | 0(0) | 1 | 1 | 0.22(0.03) | 0.14(0.01) | 1 | 1 | 0.001 | 0.001 | 1 | 1 | 0.004 | 0.004 | 1 | 1 |
| | 80 | 0.01(0) | 0(0) | 1 | 1 | 0.13(0.01) | 0.15(0.02) | 1 | 1 | 0.001 | 0.001 | 1 | 1 | 0.004 | 0.004 | 1 | 1 |
| | 100 | 0.04(0) | 0(0) | 1 | 1 | 0.14(0.01) | 0.09(0.01) | 1 | 1 | 0.001 | 0.001 | 1 | 1 | 0.004 | 0.004 | 1 | 1 |
| Kurtosis | 1 | 0.47(0.12) | 0.19(0.04) | 0.89 | 0.98 | 1.09(0.37) | 0.88(0.28) | 0.77 | 0.90 | 0.28 | 0.23 | 0.74 | 0.72 | 0.18 | 0.11 | 0.79 | 0.88 |
| | 5 | 0.16(0.03) | 0.06(0.01) | 1 | 1 | 0.63(0.18) | 0.41(0.09) | 0.96 | 0.99 | 0.04 | 0.01 | 0.94 | 0.98 | 0.03 | 0.008 | 0.97 | 0.96 |
| | 10 | 0.02(0) | 0(0) | 1 | 1 | 0.35(0.08) | 0.32(0.06) | 0.97 | 1 | 0.001 | 0.001 | 1 | 1 | 0.007 | 0.004 | 0.96 | 1 |
| | 20 | 0(0) | 0(0) | 1 | 1 | 0.20(0.03) | 0.18(0.02) | 1 | 1 | 0.001 | 0.001 | 1 | 1 | 0.004 | 0.004 | 1 | 1 |
| | 40 | 0(0) | 0(0) | 1 | 1 | 0.06(0.01) | 0.06(0) | 1 | 1 | 0.001 | 0.001 | 1 | 1 | 0.004 | 0.004 | 1 | 1 |
| | 60 | 0(0) | 0(0) | 1 | 1 | 0.05(0) | 0.04(0) | 1 | 1 | 0.001 | 0.001 | 1 | 1 | 0.004 | 0.004 | 1 | 1 |
| | 80 | 0(0) | 0(0) | 1 | 1 | 0.05(0) | 0.03(0) | 1 | 1 | 0.001 | 0.001 | 1 | 1 | 0.004 | 0.004 | 1 | 1 |
| | 100 | 0(0) | 0(0) | 1 | 1 | 0.02(0) | 0(0) | 1 | 1 | 0.001 | 0.001 | 1 | 1 | 0.004 | 0.004 | 1 | 1 |

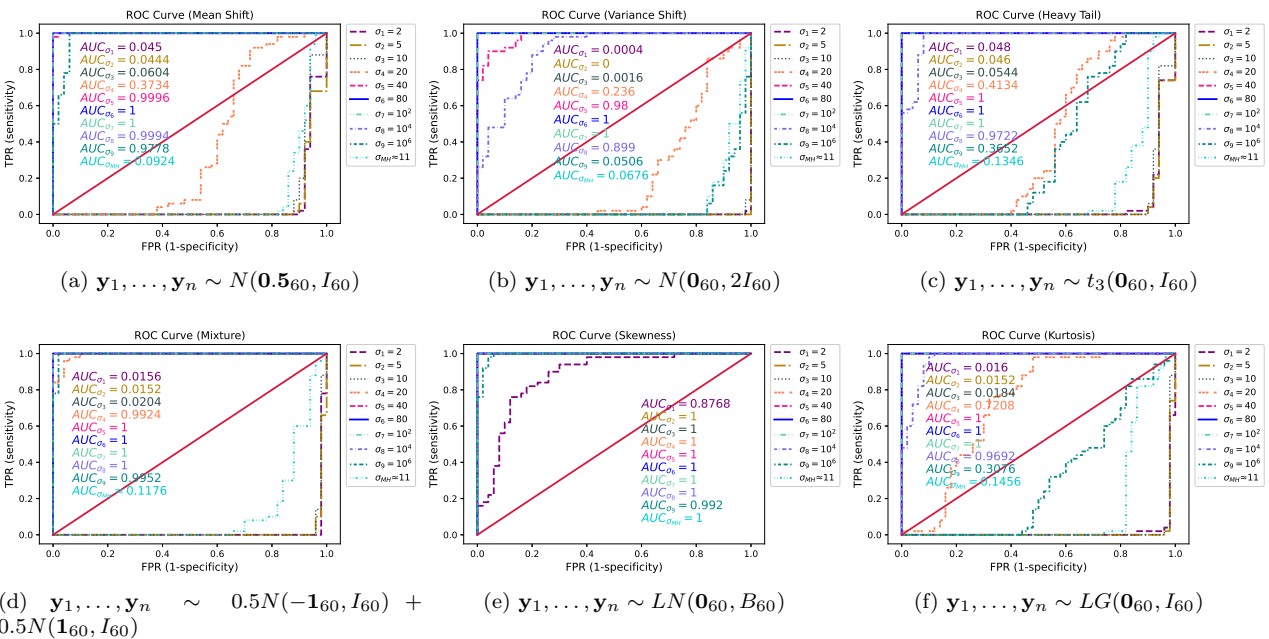

(a) $\mathbf{y}_1, \ldots, \mathbf{y}_n \sim N(\mathbf{0.5}_{60}, I_{60})$  (b) $\mathbf{y}_1, \ldots, \mathbf{y}_n \sim N(\mathbf{0}_{60}, 2I_{60})$  (c) $\mathbf{y}_1, \ldots, \mathbf{y}_n \sim t_3(\mathbf{0}_{60}, I_{60})$

(d) $\mathbf{y}_1, \ldots, \mathbf{y}_n \sim 0.5N(-\mathbf{1}_{60}, I_{60}) + 0.5N(\mathbf{1}_{60}, I_{60})$  (e) $\mathbf{y}_1, \ldots, \mathbf{y}_n \sim LN(\mathbf{0}_{60}, B_{60})$  (f) $\mathbf{y}_1, \ldots, \mathbf{y}_n \sim LG(\mathbf{0}_{60}, I_{60})$

Figure 1: The ROC curves and AUC values of the BNP-MMD test for $\mathbf{x}_1, \ldots, \mathbf{x}_n \sim N(\mathbf{0}_{60}, I_{60})$, using a range of bandwidth parameters including $\sigma = 2, 5, 10, 20, 40, 80, 10^2, 10^4, 10^6$, as well as the median heuristic $\sigma_{MH}$.

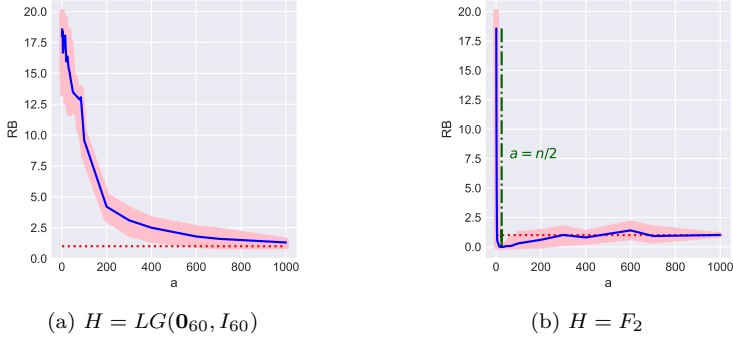

(a) $H = LG(\mathbf{0}_{60}, I_{60})$   (b) $H = F_2$

Figure 2: The solid line represents the average of the RB and the pink area represents a 95% confidence interval of the RB over the 100 samples with various choices of $H$ and $a$ for the heavy tail example. The lower and upper bounds are the 2.5% and 97.5% quantiles of the RB, respectively. The red dotted line represents $RB = 1$.

To compare the BNP and FNP tests, the $p$-values of the frequentists counterparts corresponding to each Bayesian test are presented in Table 1 using the R packages **energy**[8] and **maotai**[9]. AUC values of all tests are also given to facilitate comparison between tests. Generally, the proposed test reflects better performances than its frequentist counterparts in lower dimensions. For instance, in the variance shift example, when $d = 5$ and $n = 30$, the average of the $RB$ and its strength for the semi-BNP-MMD test are 0.55 and 0.12, respectively, which shows strong evidence to reject the null. While the average of the $p$-value corresponding to the MMD frequentist test is 0.34, which shows a failure to reject the null hypothesis. The AUC value of the semi-BNP test is also 0.99 which indicates a better ability than its frequentist counterpart with an AUC

---

[8]https://CRAN.R-project.org/package=energy
[9]https://CRAN.R-project.org/package=maotai

of 0.65. To examine the large sample property, additional results for $n = 500, 1000$ are presented in Section F.1 of the Appendix, revealing the relatively poor performance of the BNP-Energy test in comparison to other tests.

## 6.2 The Semi-BNP GAN

According to the results reported in the previous subsection, the semi-BNP estimator suggests a test that outperforms other competing tests in many scenarios. Therefore, we expect that embedding this estimator in GANs as the discriminator will accurately distinguish real and fake data. We use the database of handwritten digits with 10 modes, bone marrow biopsy histopathology, human faces, and brain MRI images to analyze the model performance. Following the design choices of Li et al. (2015), we use the Gaussian neural network for the generator with four hidden layers each having rectified linear units activation function and a sigmoid function for the output layer. For fitting a deep neural network, there are numerous methods to choose network parameters. Furthermore, we select the number of nodes in hidden layers and tuning parameters of the network using Bayesian optimization (Snoek et al., 2012). We also set mini-batch sizes to be $n_{mb} = 1,000$ and use a mixture of six Gaussian kernels corresponding to the bandwidth parameters $2, 5, 10, 20, 40,$ and $80$ to train networks discussed in this section.

### 6.2.1 MNIST Dataset (LeCun, 1998):

The MNIST dataset includes 60,000 handwritten digits of 10 numbers from 0 to 9 each having 784 ($28 \times 28$) dimensions. This dataset is split into 50000 training and 10000 testing images and is a good example to demonstrate the performance of the method in dealing with the mode collapse problem. We use the training set to train the network. A sample from the training MNIST dataset is shown in Figure 3-a. Following $r_{mb} = 40,000$ iterations, we generate samples from the trained semi-BNP GAN using Algorithm 2 from the Appendix, as depicted in Figure 3-b. The results of Li et al. (2015) are also presented by Figure 3-c as the frequentist counterpart of our semi-BNP procedure[10]. Based on these preliminary results, we can see that our generated images can, at least, replicate the results of Li et al. (2015) and in some cases produce sharper images. This result can also be deduced from the presented values of certain score functions in Section F.2 of the Appendix. On the other hand, unlike the semi-BNP test, our experimental results demonstrate that the

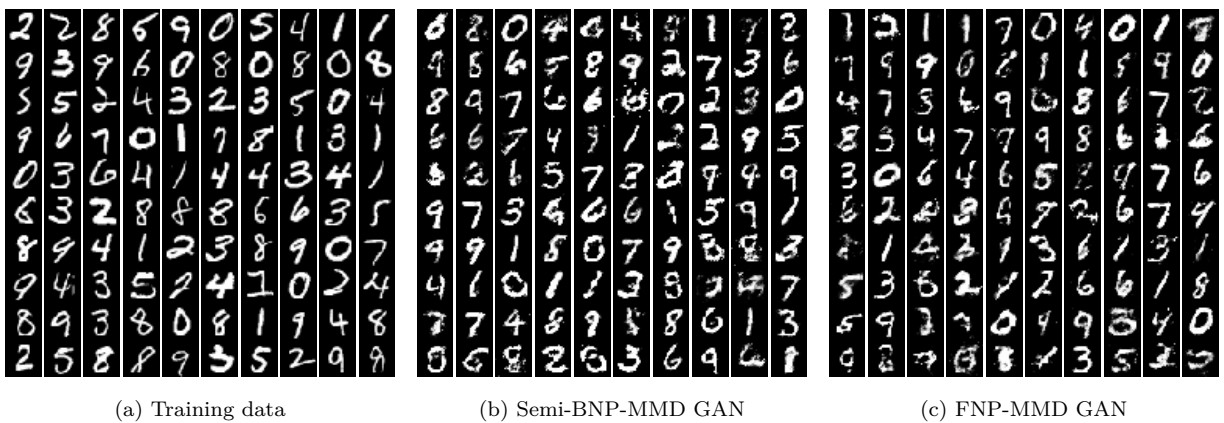

(a) Training data          (b) Semi-BNP-MMD GAN          (c) FNP-MMD GAN

Figure 3: Generated samples of sizes ($10 \times 10$) from semi-BNP-MMD and MMD-FNP GAN for the MNIST dataset using a mixture of Gaussian kernels in 40,000 iterations.

semi-BNP GAN, using a mixture of Gaussian kernels, outperforms the approach that considers only a single Gaussian kernel. To investigate this matter further, we present several samples of the trained generator using a Gaussian kernel with different values of $\sigma$, as well as the median heuristic $\sigma_{MH}$, in Figure 4. Note

---

[10]The code for the GAN proposed by Li et al. (2015) is available at `https://www.dropbox.com/s/anf9z1zyqi7379n/Generative-Moment-Matching-Networks-master.zip?file_subpath=%2FREADME.md`

that the value of $\sigma_{MH}$ is updated in each iteration, and therefore, no specific value is reported for it in this figure. While increasing the value of $\sigma$ enhances the diversity of the generated images, it is evident that the resolution of the images in Figure 4 does not reach the image quality achieved by the mixture kernel.

In contrast to using MMD kernel-based measures, it may also be interesting to consider the energy distance in learning GANs from a BNP perspective. To address this concern, we embed the two-sample BNP-energy test of Al-Labadi et al. (2022a) in training GANs as a discriminator and show the generated samples in Figure 5-a. This image clearly shows the inefficiency of the two-sample BNP test of Al-Labadi et al. (2022a) in training the generator. The main issue in this test procedure is treating $F_{G_\omega}$ as unknown distribution to place a DP prior on it which is contrary to update parameter $\omega$ in the parameterized generative neural network $G_\omega$.

One may also be interested in considering the semi-BNP-energy procedure in learning GANs in which it may be more sensible to compare the semi-BNP-MMD results. To do this, we use the energy distance instead of the MMD in Algorithm 2 in the Appendix. The results are presented in Figure 5-b and show blurry and unclear images with no variety, which reflect the inefficiency of using the energy distance compared to the MMD kernel-based measure. More experiments are given in Section F.2 of the Appendix.

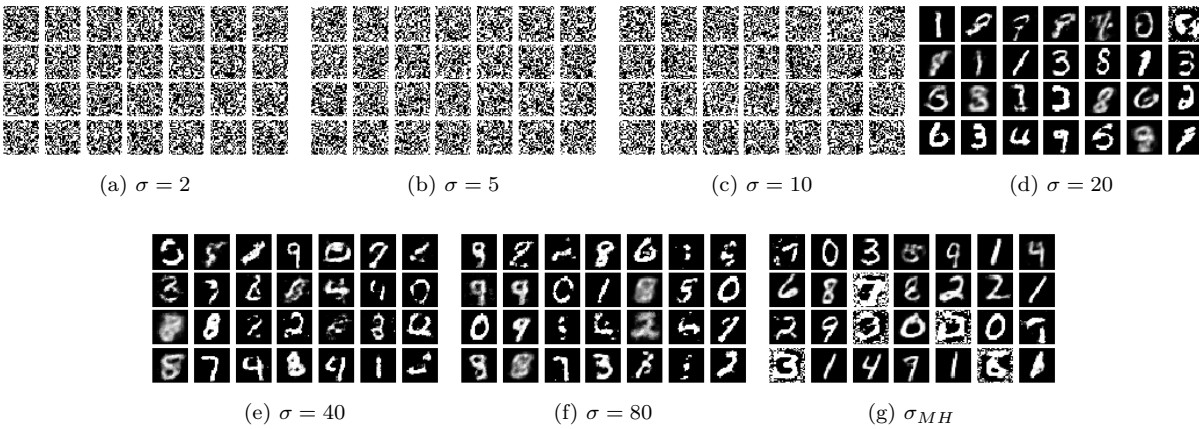

(a) $\sigma = 2$   (b) $\sigma = 5$   (c) $\sigma = 10$   (d) $\sigma = 20$

(e) $\sigma = 40$   (f) $\sigma = 80$   (g) $\sigma_{MH}$

Figure 4: Generated samples from semi-BNP-MMD for the MNIST dataset using a single Gaussian kernel with various values of bandwidth parameter $\sigma$ in 40,000 iterations.

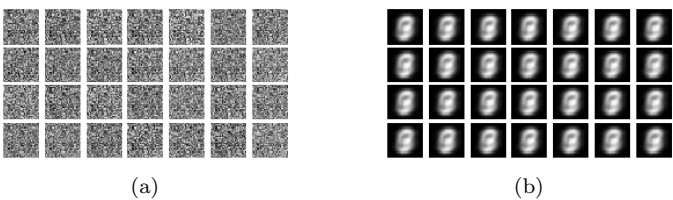

(a)   (b)

Figure 5: Generated samples from BNP-Energy GAN (a) and semi-BNP-Energy GAN (b) for the MNIST dataset in 40,000 iterations.

## 7   Conclusion

Our semi-BNP approach effectively estimates the MMD measure between an unknown distribution and an intractable parametric distribution. It outperforms frequentist counterparts and even surpasses a recent BNP competitor in certain scenarios (Al-Labadi et al., 2022a). This approach shows great potential in training GANs, where the proposed estimator serves as a discriminator, inducing a posterior distribution on the generator's parameter space. Stick-breaking representation lacks normalization terms and exhibits stochastic

decrease, making it inefficient for simulations (Zarepour & Al-Labadi, 2012). Thus, exploring alternative DP approximations for MMD estimation presents an intriguing avenue for future research. Future work will focus on generating 3D medical images to further enhance results.

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

# Appendix

# A   Technical Proofs

## A.1   Theoretical Properties of the DP Approximation given by Ishwaran & Zarepour (2002)

**Proposition 6** *For a non-negative real value $a$ and fixed probability distribution $H$, let $F_1^{pri} := F_1 \sim DP(a, H)$ and $(J_{1,N}, \ldots, J_{N,N}) \sim Dirichlet(\frac{a}{N}, \ldots, \frac{a}{N})$ be the weights in the approximation of $F^{pri}$, given by Ishwaran & Zarepour (2002). Then, as $a \to \infty$,*

i. $J_{\ell,N} \xrightarrow{a.s.} \frac{1}{N}$, for any $\ell \in \{1, \ldots, N\}$,

ii. $J_{\ell,N} J_{t,N} \xrightarrow{a.s.} \frac{1}{N^2}$, for any $\ell, t \in \{1, \ldots, N\}$, where $\ell \neq t$.

**Proof.** Recall

$$F_N^{pri} = \sum_{i=1}^N J_{i,N} \delta_{Y_i}. \tag{14}$$

Since $E_{F_1^{pri}}(J_{\ell,N}) = \frac{1}{N}$, for any $\ell \in \{1, \ldots, N\}$ and $\epsilon > 0$, Chebyshev's inequality implies

$$\Pr\{|J_{\ell,N} - 1/N| \geq \epsilon\} \leq \frac{Var(J_{\ell,N})}{\epsilon^2},$$

where, $Var_{F_1^{pri}}(J_{\ell,N}) = \frac{N-1}{N^2(a+1)}$. Assuming $a = \kappa^2 c$ for $\kappa \in \mathbb{N}$ and a fixed positive number $c$, gives

$$\Pr\{|J_{\ell,N} - 1/N| \geq \epsilon\} \leq \frac{1}{\kappa^2 c \epsilon^2}.$$

The convergence of series $\sum_{\kappa=0}^\infty \kappa^{-2}$ implies $\sum_{\kappa=0}^\infty \Pr\{|J_{\ell,N} - 1/N| \geq \epsilon\} < \infty$. By letting $a \to \infty$, the first Borel Cantelli lemma concludes $|J_{\ell,N} - 1/N| \xrightarrow{a.s.} 0$ and the result of (i) follows. To prove (ii), it is enough to show $\Pr\{\lim_{a\to\infty}(J_{\ell,N} J_{t,N}) \neq \frac{1}{N^2}\} = 0$. To prove this for the probability space $(\Omega, \mathcal{F}, \Pr)$, let

$$A = \left\{\omega \in \Omega : \lim_{a\to\infty}(J_{\ell,N}(\omega) J_{t,N}(\omega)) \neq \frac{1}{N^2}\right\}, \quad B = \left\{\omega \in \Omega : \lim_{a\to\infty}(J_{\ell,N}(\omega)) \neq \frac{1}{N}\right\},$$

$$C = \left\{\omega \in \Omega : \lim_{a\to\infty}(J_{t,N}(\omega)) \neq \frac{1}{N}\right\},$$

where, $\Pr(B)$ and $\Pr(C)$ are zero by (i). Since $A \subseteq B \cup C$, then,

$$1 - \Pr\left\{\omega \in \Omega : \lim_{a\to\infty}(J_{\ell,N}(\omega) J_{t,N}(\omega)) = \frac{1}{N^2}\right\} = \Pr(A) \leq \Pr(B) + \Pr(C) = 0,$$

which concludes the result. ∎

## A.2 Proof of Theorem 1

**Proof.** For samples $\{\mathbf{V}_\ell\}_{\ell=1}^N$ and $\{\mathbf{Y}_\ell\}_{\ell=1}^m$, respectively, from $H$ and $F_2$, the triangle inequality implies

$$\left|\text{MMD}_{\text{BNP}}^2(F_{1,N}^{pri}, F_{2,m}) - \text{MMD}^2(H_N, F_{2,m})\right| \leq K\left\{\sum_{\ell,t=1}^N \left|J_{\ell,N} J_{t,N} - \frac{1}{N^2}\right|\right.$$

$$\left. + \frac{2}{m}\sum_{\ell=1}^N \sum_{t=1}^m \left|J_{\ell,N} - \frac{1}{N}\right|\right\}.$$

By Proposition 6, which provides some theoretical properties of the DP approximation given in (14), the right-hand side of the above inequality converges almost surely to 0 as $a \to \infty$ for fixed $N$. This convergence immediately concludes the proof of (i). To prove (ii), since $(J_{1,N}, \ldots, J_{N,N}) \sim \text{Dirichlet}(\frac{a}{N}, \ldots, \frac{a}{N})$, $E_{F_1^{pri}}(J_{\ell,N}) = \frac{1}{N}$ and

$$E_{F_1^{pri}}(J_{\ell,N} J_{t,N}) = \begin{cases} \dfrac{a}{(a+1)N^2} & \text{if } \ell \neq t, \\ \dfrac{a+N}{(a+1)N^2} & \text{if } \ell = t. \end{cases}$$

Applying these properties in definition of $\text{MMD}^2_{\text{BNP}}(F^{pri}_{1,N}, F_{2,m})$ results in

$$E_{F^{pri}_1}(\text{MMD}^2_{\text{BNP}}(F^{pri}_{1,N}, F_{2,m})|\mathbf{V}_{1:N}) = \sum_{\ell=1}^{N}\sum_{t\neq\ell}^{N} \frac{ak(\mathbf{V}_\ell, \mathbf{V}_t)}{(a+1)N^2} + \sum_{\ell=1}^{N}\sum_{t=\ell}^{N} \frac{(a+N)k(\mathbf{V}_\ell, \mathbf{V}_t)}{(a+1)N^2}$$

$$- \frac{2}{Nm}\sum_{\ell=1}^{N}\sum_{t=1}^{m} k(\mathbf{V}_\ell, \mathbf{Y}_t) + \frac{1}{m^2}\sum_{\ell,t=1}^{m} k(\mathbf{Y}_\ell, \mathbf{Y}_t). \tag{15}$$

Now, it is sufficient to compute the following conditional expectation,

$$E(\text{MMD}^2_{\text{BNP}}(F^{pri}_{1,N}, F_{2,m})) = E_{H,F_2}(E_{F^{pri}_1}(\text{MMD}^2_{\text{BNP}}(F^{pri}_{1,N}, F_{2,m})|\mathbf{V}_{1:N})). \tag{16}$$

Since sets $\{V_i\}_{i=1}^{N}$ and $\{Y_i\}_{i=1}^{m}$ include i.i.d. random variables, separately, replacing equation 15 in expectation (16) implies:

$$(16) = \frac{a(N-1)}{(a+1)N}E_H[k(\mathbf{V}_1, \mathbf{V}_2)] + \frac{a+N}{(a+1)N}E_H[k(\mathbf{V}_1, \mathbf{V}_1)] - 2E_{H,F_2}[k(\mathbf{V}_1, \mathbf{Y}_1)]$$

$$+ \frac{m-1}{m}E_{F_2}[k(\mathbf{Y}_1, \mathbf{Y}_2)] + \frac{1}{m}E_{F_2}[k(\mathbf{Y}_1, \mathbf{Y}_1)]. \tag{17}$$

The proof of (ii) is concluded by letting $a \to \infty$, $N \to \infty$, and $m \to \infty$ in the above equation. Lastly, since $\frac{1}{m} < 1$, $\frac{m-1}{m} < 1$, $\frac{a(N-1)}{(a+1)N} < 1$, and $\frac{a+N}{(a+1)N} < 2$, then, for any $N, m \in \mathbb{N}$ and $a \in \mathbb{R}^+$,

$$(17) < E_H[k(\mathbf{V}_1, \mathbf{V}_2)] - 2E_{H,F_2}[k(\mathbf{V}_1, \mathbf{Y}_1)] + E_{F_2}[k(\mathbf{Y}_1, \mathbf{Y}_2)] + 3K,$$

which concludes the proof of (iii). ∎

### A.3 Proof of Theorem 2

**Proof.** Applying triangular inequality implies

$$\left|\text{MMD}^2_{\text{BNP}}(F^{pos}_{1,N}, F_{2,m}) - \text{MMD}^2(H_N, F_{2,m})\right| \leq \sum_{\ell,t=1}^{N} \left|J^*_{\ell,N}J^*_{t,N}k(\mathbf{V}^*_\ell, \mathbf{V}^*_t) - \frac{1}{N^2}k(\mathbf{V}_\ell, \mathbf{V}_t)\right|$$

$$+ \frac{2}{m}\sum_{\ell=1}^{N}\sum_{t=1}^{m} \left|J^*_{\ell,N}k(\mathbf{V}^*_\ell, \mathbf{Y}_t) - \frac{1}{N}k(\mathbf{V}_\ell, \mathbf{Y}_t)\right|, \tag{18}$$

where, samples $\{\mathbf{V}^*_\ell\}_{\ell=1}^{N}$ and $\{\mathbf{Y}_\ell\}_{\ell=1}^{m}$ are generated from $H^*$ and $F_2$, respectively. Similar to Proposition 6, it can be shown that $J^*_{\ell,N} \to 1/N$ and $J^*_{\ell,N}J^*_{t,N} \to 1/N^2$, as $a \to \infty$, using conjugacy property of DP. On the other hand, since $H^* \to H$ as $a \to \infty$, the chance of sampling from $H$ and $F_{1,n}$ tends, respectively, to one and zero, which implies $V^*_i \to V_i$, where $V_i \sim H$, for $i = 1, 2$. Applying the continuous mapping theorem implies $k(\mathbf{V}^*_1, \mathbf{V}^*_2) \to k(\mathbf{V}_l, \mathbf{V}_t)$ and $k(\mathbf{V}^*_l, \mathbf{Y}_t) \to k(\mathbf{V}_l, \mathbf{Y}_t)$, which completes the proof of (i)(a). To prove (i)(b), it follows from the proof of Theorem 1:

$$E(\text{MMD}^2_{\text{BNP}}(F^{pos}_{1,N}, F_{2,m})) = h_1(a, n, N)E_{H^*}[k(\mathbf{V}^*_1, \mathbf{V}^*_2)] + h_2(a, n, N)E_{H^*}[k(\mathbf{V}^*_1, \mathbf{V}^*_1)]$$

$$- 2E_{H^*,F_2}[k(\mathbf{V}^*_1, \mathbf{Y}_1)] + \frac{m-1}{m}E_{F_2}[k(\mathbf{Y}_1, \mathbf{Y}_2)]$$

$$+ \frac{1}{m}E_{F_2}[k(\mathbf{Y}_1, \mathbf{Y}_1)], \tag{19}$$

where $h_1(a, n, N) = \frac{(a+n)(N-1)}{(a+n+1)N}$ and $h_2(a, n, N) = \frac{a+n+N}{(a+n+1)N}$. Since $k(\cdot, \cdot)$ is bounded above by $K$, the dominated convergence theorem implies $E_{H^*}[k(\mathbf{V}^*_1, \mathbf{V}^*_2)] \to E_H[k(\mathbf{V}_1, \mathbf{V}_2)]$ and $E_{H^*,F_2}[k(\mathbf{V}^*_1, \mathbf{Y}_1)] \to E_{H,F_2}[k(\mathbf{V}_1, \mathbf{Y}_1)]$. Since $h_1(a, n, N) \to 1$ and $h_2(a, n, N) \to 0$ as $a \to \infty$, $N \to \infty$; and, $m/(m-1) \to 1$ and $1/m \to 0$, as $m \to \infty$, the results follow.

To prove (ii)(a) and (ii)(b), $F_{1,n} \to F_1$, and then $H^* \to F_1$ as $n \to \infty$ by the Glivenko-Cantelli theorem. It indicates that the probability of sampling from $H$ and $F_{1,n}$ tends, respectively, to zero and one. Therefore, $V_i^* \to X_i$ as $n \to \infty$, where $X_i \sim F_1$, for $i = 1, 2$. The proof of (ii)(a) is completed with the same strategy as the proof of (i)(a) by letting $n \to \infty$ in (18). The proof of (ii)(b) is also concluded with a similar argument that in (i)(b), when $n \to \infty$ in (19). ∎

### A.4 Proof of Corollary 3

**Proof.** The proofs are immediately followed by Theorem 1 and Theorem 2. ∎

### A.5 Proof of Lemma 4

**Proof.** The proof of Lemma 4(i) relies on the proof given in Dellaporta et al. (2022, Theorem 9) which is expanded for infinite stick-breaking representation, while we consider the finite DP approximation given in (14). By employing a similar technique as in the previously mentioned theorem, we have

$$
\begin{aligned}
E\left(\mathrm{MMD}(F, F_{G_{\boldsymbol{\omega}^*}})\right) &= E_F\left(E_{F^{pos}}\mathrm{MMD}(F, F_{G_{\boldsymbol{\omega}^*}})|\mathbf{X}_{1:n}\right) \\
&\leq \min_{\boldsymbol{\omega} \in \mathcal{W}} \mathrm{MMD}(F, F_{G_{\boldsymbol{\omega}}}) + 2E_F\left(\mathrm{MMD}(F_n, F)\right) + 2E_{F^{pos}}\left(\mathrm{MMD}(F_N^{pos}, H^*)\right) \\
&\quad + 2E_F\left(E_{H^*}(\mathrm{MMD}(F_n, H^*)|\mathbf{X}_{1:n})\right).
\end{aligned}
$$

Building on the results of Dellaporta et al. (2022, Lemma 7), we can establish that

$$
E_{F^{pos}}\left(\mathrm{MMD}^2(F_N^{pos}, H^*)\right) \leq \sum_{\ell=1}^N E_{F^{pos}}[J_{\ell,N}^{*2}]E_{H^*}[k(\mathbf{V}_\ell^*, \mathbf{V}_\ell^*)] \leq \frac{(a+n+N)K}{(a+n+1)N},
$$

where the right-hand side of the above inequality follows from the fact that $k(\cdot, \cdot) \leq K$ and $E_{F^{pos}}[J_{\ell,N}^{*2}] = \frac{a+n+N}{(a+n+1)N^2}$. Now, the Jensen's inequality implies

$$
E_{F^{pos}}\left(\mathrm{MMD}(F_N^{pos}, H^*)\right) \leq \sqrt{\frac{(a+n+N)K}{(a+n+1)N}}.
$$

On the other hand, Chérief-Abdellatif & Alquier (2022, Lemma 7.1) and Dellaporta et al. (2022, Lemma 8), respectively, imply that

$$
E_F\left(\mathrm{MMD}(F_n, F)\right) \leq \frac{K}{\sqrt{n}}, E_F\left(E_{H^*}(\mathrm{MMD}(F_n, H^*)|\mathbf{X}_{1:n})\right) \leq \frac{2aK}{a+n},
$$

which concludes the proof of (i). To establish (ii), we adopt the approach used in the proof of Dellaporta et al. (2022, Corollary 5). Initially, we employ Chérief-Abdellatif & Alquier (2022, Lemma 3.3) to bound $\mathrm{MMD}(F_0, F_{G_{\boldsymbol{\omega}^*}})$ by $2\epsilon + \mathrm{MMD}(F, F_{G_{\boldsymbol{\omega}^*}})$, resulting in:

$$
E\left(\mathrm{MMD}(F_0, F_{G_{\boldsymbol{\omega}^*}})\right) \leq 2\epsilon + E\left(\mathrm{MMD}(F, F_{G_{\boldsymbol{\omega}^*}})\right).
$$

Applying the result in (i) to the right-hand side of the above inequality implies:

$$
E\left(\mathrm{MMD}(F_0, F_{G_{\boldsymbol{\omega}^*}})\right) \leq 2\epsilon + \min_{\boldsymbol{\omega} \in \mathcal{W}} \mathrm{MMD}(F, F_{G_{\boldsymbol{\omega}}}) + \frac{2K}{\sqrt{n}} + \frac{4aK}{a+n} + 2\sqrt{\frac{(a+n+N)K}{(a+n+1)N}}.
$$

Finally, we employ Chérief-Abdellatif & Alquier (2022, Lemma 3.3) once again, but this time to bound $\mathrm{MMD}(F, F_{G_{\boldsymbol{\omega}}})$ by $2\epsilon + \mathrm{MMD}(F_0, F_{G_{\boldsymbol{\omega}}})$ for any $\boldsymbol{\omega} \in \mathcal{W}$, thereby completing the proof of (ii). ∎

## A.6 Proof of Lemma 5

**Proof.** Let $\mathcal{L}_{\text{BNP}}(\boldsymbol{\omega}) = \text{MMD}(F_N^{pos}, F_{G_{\boldsymbol{\omega}}})$, $\mathcal{L}_{n,m}(\boldsymbol{\omega}) = \text{MMD}(F_n, F_{G_{\boldsymbol{\omega}},m})$, and $\mathcal{L}(\boldsymbol{\omega}) = \text{MMD}(F, F_{G_{\boldsymbol{\omega}}})$. Then, for $\boldsymbol{\omega}^* \in \mathcal{W}$, Gretton et al. (2012a, Theorem 7) implies

$$\Pr\left(|\mathcal{L}_{n,m}(\boldsymbol{\omega}^*) - \mathcal{L}(\boldsymbol{\omega}^*)| > h(n,m,K,\epsilon)\right) < 2\exp\frac{-\epsilon^2 nm}{2K(n+m)}. \tag{20}$$

Hence, with a probability at least $1 - 2\exp\frac{-\epsilon^2 nm}{2K(n+m)}$,

$$|\mathcal{L}_{n,m}(\boldsymbol{\omega}^*) - \mathcal{L}(\boldsymbol{\omega}^*)| \leq h(n,m,K,\epsilon). \tag{21}$$

On the other hand, the triangle inequality implies

$$|\mathcal{L}_{\text{BNP}}(\boldsymbol{\omega}^*) - \mathcal{L}(\boldsymbol{\omega}')| \leq |\mathcal{L}_{n,m}(\boldsymbol{\omega}^*) - \mathcal{L}(\boldsymbol{\omega}^*)| + |\mathcal{L}_{\text{BNP}}(\boldsymbol{\omega}^*) - \mathcal{L}_{n,m}(\boldsymbol{\omega}^*)| + |\mathcal{L}(\boldsymbol{\omega}^*) - \mathcal{L}(\boldsymbol{\omega}')|. \tag{22}$$

Finally, the proof of (i) is concluded by considering inequality 21 in equation 22. To prove (ii), Markov's inequality implies

$$\Pr\left(\text{MMD}(F, F_{G_{\boldsymbol{\omega}^*}}) \geq \epsilon\right) \leq \frac{E\left(\text{MMD}(F, F_{G_{\boldsymbol{\omega}^*}})\right)}{\epsilon}.$$

The result follows by substituting the bounds from Lemma 4(i) into the right-hand side of the above inequality. ∎

# B Computational Algorithms

## B.1 Implementing the Semi-BNP GOF Kernel-based Test

---

**Algorithm 1** Pseudocode of semi-BNP two-sample MMD kernel test

---

1: Initialize $a$, $\ell$, $M$, $i_0$, and $\epsilon$ in equation 4 to determine $N$. Consider an observed sample $\mathbf{x}_{1:n}$ from $F_1$.
2: $H \leftarrow F_2$
   STEP 1: Computing the BNP MMD
3: **for** $r \leftarrow 0$ to $\ell$ **do**
4:     Generate an approximate sample of $F_1 \sim DP(a, H)$ by using $\sum_{i=1}^N J_{i,N}\delta_{\mathbf{V}_i}$, where $\{J_{i,N}\}_{i=1}^N \sim$ Dirichlet$(\frac{a}{N}, \cdots, \frac{a}{N})$, and $\{\mathbf{V}_i\}_{i=1}^N \sim H$.
5:     Generate an approximate sample of $F_1|\mathbf{x}_{1:n} \sim DP(a+n, H^*)$ by using $\sum_{i=1}^N J_{i,N}^*\delta_{\mathbf{V}_i^*}$, where $\{J_{i,N}^*\}_{i=1}^N \sim$ Dirichlet$(\frac{a+n}{N}, \cdots, \frac{a+n}{N})$, and $\{\mathbf{V}_i^*\}_{i=1}^N \sim H^*$.
6:     Use samples generated in steps 4 and 5 to compute
   $\text{MMD}_{\text{BNP}}^2(F_{1,N}^{pri}, F_{2,m})$ and $\text{MMD}_{\text{BNP}}^2(F_{1,N}^{pos}, F_{2,m})$, respectively.
7: **end for**
8: **return** $\{\text{MMD}_{\text{BNP}_r}^2(F_{1,N}^{pri}, F_{2,m})\}_{r=1}^\ell$ and $\{\text{MMD}_{\text{BNP}_r}^2(F_{1,N}^{pos}, F_{2,m})\}_{r=1}^\ell$
   STEP 2: Estimating RB and Str
9: $\widehat{\Pi}_{\text{MMD}^2}(\cdot|\mathbf{x}_{1:n}) \leftarrow ECDF(\{\text{MMD}_{\text{BNP}_r}^2(F_{1,N}^{pos}, F_{2,m})\}_{r=1}^\ell)$       ▷ The ECDF of posterior-based MMD
10: $\widehat{\Pi}_{\text{MMD}^2}(\cdot) \leftarrow ECDF(\{\text{MMD}_{\text{BNP}_r}^2(F_{1,N}^{pri}, F_{2,m})\}_{r=1}^\ell)$       ▷ The ECDF of prior-based MMD
11: $\widehat{d}_{i_0/M} \leftarrow quantile(\{\text{MMD}_{\text{BNP}_r}^2(F_{1,N}^{pri}, F_{2,m})\}_{r=1}^\ell, i_0/M)$     ▷ The estimation of the $i_0/M$-th quantile of
   $\text{MMD}_{\text{BNP}}^2(F_{1,N}^{pri}, F_{2,m})$
12: $\widehat{RB}_{\text{MMD}^2}(0|\mathbf{x}_{1:n}) \leftarrow \frac{\widehat{\Pi}_{\text{MMD}^2}(\widehat{d}_{i_0/M}|\mathbf{x}_{1:n})}{\widehat{\Pi}_{\text{MMD}^2}(\widehat{d}_{i_0/M})}$
13: $\widehat{Str} \leftarrow 0$
14: **for** $i \leftarrow 0$ to $M-1$ **do**
15:     $\widehat{d}_{i/M} \leftarrow quantile(\{\text{MMD}_{\text{BNP}_r}^2(F_{1,N}^{pri}, F_{2,m})\}_{r=1}^\ell, i/M)$
16:     $\widehat{d}_{(i+1)/M} \leftarrow quantile(\{\text{MMD}_{\text{BNP}_r}^2(F_{1,N}^{pri}, F_{2,m})\}_{r=1}^\ell, (i+1)/M)$
17:     $\widehat{RB}_{\text{MMD}^2}(\widehat{d}_{i/M}|\mathbf{x}_{1:n}) \leftarrow \frac{\widehat{\Pi}_{\text{MMD}^2}(\widehat{d}_{(i+1)/M}|\mathbf{x}_{1:n}) - \widehat{\Pi}_{\text{MMD}^2}(\widehat{d}_{i/M}|\mathbf{x}_{1:n})}{\widehat{\Pi}_{\text{MMD}^2}(\widehat{d}_{(i+1)/M}) - \widehat{\Pi}_{\text{MMD}^2}(\widehat{d}_{i/M})}$

18:     **if** $\widehat{RB}_{\mathrm{MMD}^2}(\widehat{d}_{i/M}|\mathbf{x}_{1:n}) \leq \widehat{RB}_{\mathrm{MMD}^2}(0|\mathbf{x}_{1:n})$ **then**

19:       $\widehat{Str}(0|\mathbf{x}_{1:n}) \leftarrow \widehat{Str} + [\widehat{\Pi}_{\mathrm{MMD}^2}(\widehat{d}_{(i+1)/M}|\mathbf{x}_{1:n}) - \widehat{\Pi}_{\mathrm{MMD}^2}(\widehat{d}_{i/M}|\mathbf{x}_{1:n})]$

20:     **end if**

21: **end for**

22: **return** $\widehat{RB}_{\mathrm{MMD}^2}, \widehat{Str}$

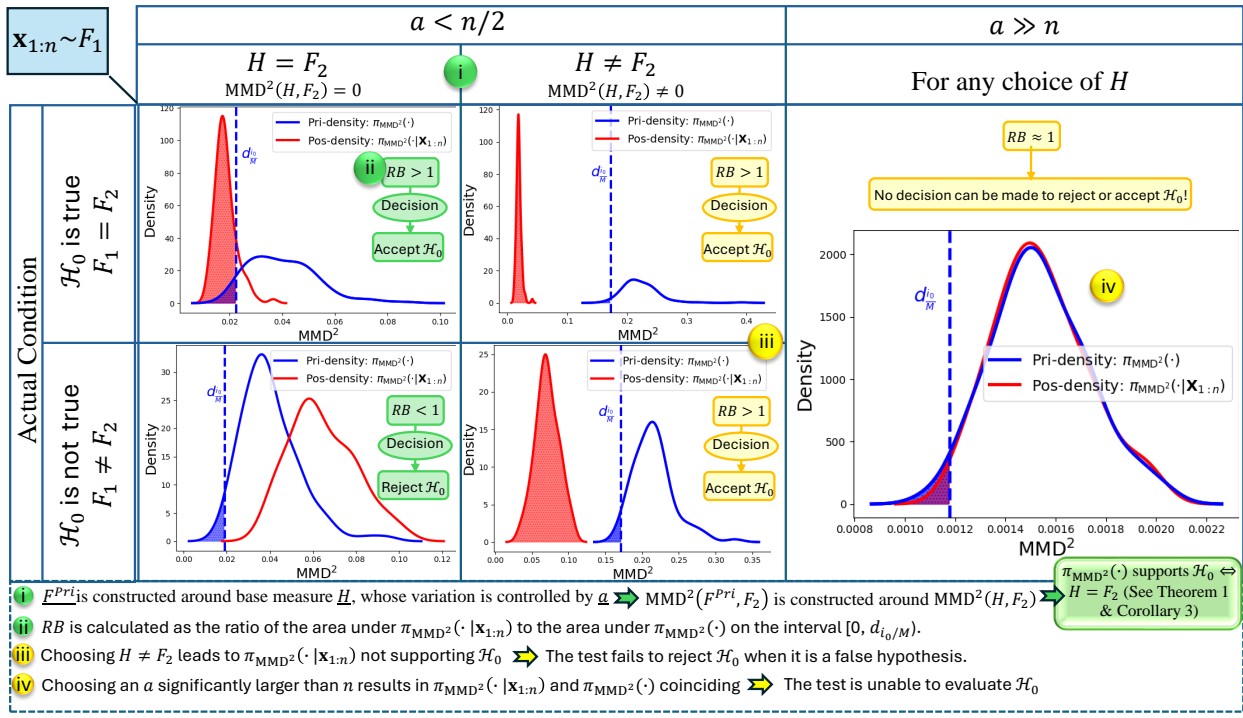

Figure 6: General diagram of the BNP-GOF test. The prior density of MMD$^2$ versus the posterior density after observing $\mathbf{x}_{1:n}$ from distribution $F_1$ under all possible conditions for both true and false null hypothesis. Here, we consider $F_1 = F_2 = N(\mathbf{0}_{60}, I_{60})$ as the true null hypothesis, while $F_1 = t_3(\mathbf{0}_{60}, I_{60})$ and $F_2 = N(\mathbf{0}_{60}, I_{60})$ for the false null hypothesis to plot densities.

## B.2   Training the Semi-BNP GAN

**Algorithm 2** Pseudocode of training a GAN using the semi-BNP approach

1: Set $a = 10^{-6}$ to employ a non-informative prior leading DP posterior $DP(n, F_n)$.

2: Initialize $\epsilon$ in equation 4 to determine $N$ using conjugacy property of DP.

3: $r_{mn} \leftarrow$ Number of training iteration, $n_{mb} \leftarrow$ Mini-batch size

4: $\boldsymbol{\omega}_0 \leftarrow$ An initial parameter for generator $G_{\boldsymbol{\omega}}$, $\{\mathbf{x}_\ell\}_{\ell=1}^n \leftarrow$ real dataset

5: **for** $i \leftarrow 0$ to $r_{mb}$ **do**

6:     Generate a random sample $\{\mathbf{x}_\ell^{mb}\}_{\ell=1}^{n_{mb}}$ from real dataset $\{\mathbf{x}_\ell\}_{\ell=1}^n$

7:     Generate a sample of noise vector $\{\mathbf{u}_\ell\}_{\ell=1}^{n_{mb}}$ from uniform distribution $U(-1, 1)$

8:     Generate a sample from $F_{G_{\boldsymbol{\omega}_i}}$, distribution of $G_{\boldsymbol{\omega}_i}$, as $\{\mathbf{y}_\ell = G_{\boldsymbol{\omega}_i}(\mathbf{u}_\ell)\}_{\ell=1}^{n_{mb}}$

9:     Generate a sample of size $N$ from $F^{pos} = F|\{\mathbf{x}_\ell^{mb}\}_{\ell=1}^{n_{mb}}$ using $\sum_{i=1}^N J_{i,N}^* \delta_{\mathbf{v}_i^*}$ by replacing $F_1$ by $F$, and $\{\mathbf{x}_\ell^{mb}\}_{\ell=1}^{n_{mb}}$ by $\mathbf{x}$ in step (4) of Algorithm 1.

10:     Use generated samples in steps 9 and 10 to compute $\mathrm{MMD}_{\mathrm{BNP}}^2(F_N^{pos}, F_{G_{\boldsymbol{\omega}_i}, N})$.

11:     Compute the gradient:

$$\frac{\partial \mathrm{MMD}_{\mathrm{BNP}}(F_N^{pos}, F_{G_{\boldsymbol{\omega}_i}, m})}{\partial \boldsymbol{\omega}_i} = \frac{1}{2\sqrt{\mathrm{MMD}_{\mathrm{BNP}}^2(F_N^{pos}, F_{G_{\boldsymbol{\omega}}, m})}} \frac{\partial \mathrm{MMD}_{\mathrm{BNP}}^2(F_N^{pos}, F_{G_{\boldsymbol{\omega}}, m})}{\partial \boldsymbol{\omega}}.$$

12:     Use backpropagation for calculating partial derivatives $\frac{\partial \mathbf{G}_{\boldsymbol{\omega}_i}(\mathbf{u}_\ell)}{\partial \boldsymbol{\omega}_i}$ in the previous step to update parameter $\boldsymbol{\omega}_i$.
13: **end for**
14: **return** $\boldsymbol{\omega}^*$         ▷ An optimized parameter for $G_{\boldsymbol{\omega}}$ that minimizes the cost function.

### B.3   Hypothesis Testing Evaluation

**Algorithm 3** Pseudocode of plotting ROC and computing AUC in semi-BNP test

1: Initialize $a$, $N$, $\ell$, and $M$.
2: Set the number of repeating experiments: $r \leftarrow 100$
3: $RB^\dagger|\mathcal{H}_0 \leftarrow$ Compute $RB$ for $r$ samples of sizes $n$ generated under the null hypothesis.
4: $RB|\mathcal{H}_1 \leftarrow$ Compute $RB$ for $r$ samples of sizes $n$ generated under the alternative hypothesis.
5: $T \leftarrow$ A sequence of numbers between 0 to $20^\ddagger$ with length $L$.    ▷ The discrimination threshold for the semi-BNP test.
6: $TP \leftarrow$ A vector whose each component represents the number of components of the vector $RB|\mathcal{H}_1$ which is less than each component of $T$.
7: $FN \leftarrow$ A vector whose each component represents the number of components of the vector $RB|\mathcal{H}_1$ which is greater than each component of $T$.
8: $FP \leftarrow$ A vector whose each component represents the number of components of the vector $RB|\mathcal{H}_0$ which is less than each component of $T$.
9: $TN \leftarrow$ A vector whose each component represents the number of components of the vector $RB|\mathcal{H}_0$ which is greater than each component of $T$.
10: Compute the confusion matrix as:

$$\begin{pmatrix} TNR := \frac{TN}{TN+FP} & FNR := \frac{FN}{FN+TP} \\ \text{(1-Type I error)} & \text{(Type II error)} \\ \\ FPR := \frac{FP}{FP+TN} & TPR := \frac{TP}{TP+FN} \\ \text{(Type I error)} & \text{(1-Type II error)} \end{pmatrix}.$$

11: ROC $\leftarrow$ Drawing a linear plot of $TPR$ against $FPR$.
12: AUC $\leftarrow$ Computing the area under the ROC.
13: **return** ROC and AUC.

$^\dagger$ It should be changed to the $p$-value in the FNP test.
$^\ddagger$ It should be changed to 1 in the FNP test.

## C   Relative Belief Ratio: A Bayesian Measure of Evidence

The RB ratio (Evans, 2015) is a form of Bayesian evidence in hypothesis testing problems and has shown excellent performance in many statistical hypothesis testing procedures (Al-Labadi et al., 2022a; 2021a; 2022b). The RB ratio is defined by the ratio of the posterior density to the prior density at a particular parameter of interest in the population distribution whose correctness is under investigation. Precisely, for a statistical model $(\mathfrak{X}, \mathcal{F})$ with $\mathcal{F} = \{f_\theta : \theta \in \Theta\}$, let $\pi$ be a prior on the parameter space $\Theta$ and $\pi(\theta \,|\, \mathbf{x}_{1:n})$ be the posterior distribution of $\theta$ after observing the data $\mathbf{x}_{1:n} = (\mathbf{x}_1, \ldots, \mathbf{x}_n)$. Consider a parameter of interest, $\psi = \Psi(\theta)$ such that $\Psi$ satisfies regularity conditions so that the prior density $\pi_\Psi$ and the posterior density $\pi_\Psi(\cdot \,|\, \mathbf{x}_{1:n})$ of $\psi$ exist with respect to some support measure on the range space for $\Psi$. When $\pi_\Psi$ and $\pi_\Psi(\cdot \,|\, \mathbf{x}_{1:n})$ are continuous at $\psi$, the RB ratio for a value $\psi$ is given by

$$RB_\Psi(\psi \,|\, \mathbf{x}_{1:n}) = \pi_\Psi(\psi \,|\, \mathbf{x}_{1:n}) / \pi_\Psi(\psi).$$

Otherwise for a sequence $N_\delta(\psi)$, the neighborhoods of $\psi$ that converge nicely to $\psi$ as $\delta \to 0$, the RB ratio is defined by $RB_\Psi(\psi \,|\, \mathbf{x}_{1:n}) = \lim_{\delta \to 0} \Pi_\Psi(N_\delta(\psi) \,|\, \mathbf{x}_{1:n}) / \Pi_\Psi(N_\delta(\psi))$, where $\Pi_\Psi$ and $\Pi_\Psi(\cdot \,|\, \mathbf{x}_{1:n})$ are the marginal prior and the marginal posterior probability measures, respectively.

Note that $RB_\Psi(\psi \,|\, \mathbf{x}_{1:n})$ measures the change in the belief of $\psi$ being the true value *a priori* to *a posteriori*. Therefore, it is a measure of evidence. If $RB_\Psi(\psi \,|\, \mathbf{x}_{1:n}) > 1$, then the probability of $\psi$ being the true value from a priori to a posteriori is increased, consequently there is evidence based on the data that $\psi$ is the true value. If $RB_\Psi(\psi \,|\, \mathbf{x}_{1:n}) < 1$, then the probability of $\psi$ being the true value from a priori to a posteriori is decreased. Accordingly, there is evidence against based on the data that $\psi$ being the true value. For the

case $RB_\Psi(\psi \,|\, \mathbf{x}_{1:n}) = 1$ there is no evidence in either direction. For the null hypothesis $\mathcal{H}_0 : \Psi(\theta) = \psi_0$, it is obvious $RB_\Psi(\psi_0 \,|\, \mathbf{x}_{1:n})$ measures the evidence in favor of or against $\mathcal{H}_0$. In this scenario where evidence for the null hypothesis is plausible, the frequentist notion of controlling the probability of falsely rejecting $\mathcal{H}_0$ (type I error) does not apply.

The possibility of calibrating RB ratios is a desirable feature that makes it attractive in hypothesis testing problems. After computing the RB ratio, it is very critical to know whether the obtained value represents strong or weak evidence for or against $\mathcal{H}_0$. A typical calibration of $RB_\Psi(\psi_0 \,|\, \mathbf{x}_{1:n})$ is given by the *strength of evidence*

$$Str_\Psi(\psi_0 \,|\, \mathbf{x}_{1:n}) = \Pi_\Psi \left[ RB_\Psi(\psi \,|\, \mathbf{x}_{1:n}) \leq RB_\Psi(\psi_0 \,|\, \mathbf{x}_{1:n}) \,|\, \mathbf{x}_{1:n} \right]. \tag{23}$$

The value of equation 23 indicates that the posterior probability that the true value of $\psi$ has a RB ratio no greater than that of the hypothesized value $\psi_0$. When $RB_\Psi(\psi_0 \,|\, \mathbf{x}_{1:n}) < 1$, there is evidence against $\psi_0$, then a small value of (23) indicates strong evidence against $\psi_0$ because the posterior probability of the true value having RB ratio bigger is large. On the other hand, a large value for (23) indicates weak evidence against $\psi_0$. Similarly, when $RB_\Psi(\psi_0 \,|\, \mathbf{x}_{1:n}) > 1$, there is evidence in favor of $\psi_0$, then a small value of (23) indicates weak evidence in favor of $\psi_0$, while a large value of (23) indicates strong evidence in favor of $\psi_0$.

The RB can be considered as a strong alternative to the Bayes factor (BF) criteria. The BF is defined as the ratio of the marginal likelihood of data under the null hypothesis to the alternative hypothesis in Bayesian hypothesis testing problems. However, computing the BF often involves intractable calculations of marginal likelihoods, which typically require computationally burdensome methods such as MCMC. The tests proposed by Holmes et al. (2015) and Borgwardt & Ghahramani (2009) are two examples of BNP tests that utilize marginal likelihood computation, and their practical usage in high-dimensional statistics is low due to this computational issue.

On the other hand, the construction of tests using the BF relies on assigning a prior $\pi_0$ to the null hypothesis $\mathcal{H}_0$, a prior $\pi_1$ to the alternative hypothesis $\mathcal{H}_1$, and a discrete probability mass $p_0$ for $\mathcal{H}_0$. However, practitioners often face challenges in eliciting these prior components within the overall prior $\pi = p_0\pi_0 + (1 - p_0)\pi_1$. Another concern of using BFs is their calibration to indicate whether weak or strong evidence is attained. For example, Jeffreys (1961) and Kass & Raftery (1995) proposed similar rules to calibrate BFs but García-Donato & Chen (2005) pointed out that such rules are inappropriate to calibrate BFs as they ignore the randomness of the data and, again, lead to improper inference[11].

## D    Radial Basis Function Kernels Family

The construction of MMD-based procedures is proposed based on considering a kernel function with feature space corresponding to a universal RKHS. The radial basis function (RBF) kernel is the most well-known kernel family satisfying the above situation. For two vectors $\mathbf{X}, \mathbf{Y} \in \mathbb{R}^d$, the RBF kernel is represented by

$$k(\mathbf{X}, \mathbf{Y}) = h(||\mathbf{X} - \mathbf{Y}||/\sigma),$$

where, $h$ is a function from the positive real numbers $\mathbb{R}^+$ to $\mathbb{R}^+$, $|| \cdot ||$ represents the $L^2$-norm, and $\sigma$ is the bandwidth parameter that indicates the kernel size. There are many functions assigned to $h$, for example, the Gaussian, exponential, rational quadratic kernels, and Matern, represented by

$$h_1(x) = \exp\left(-\frac{x^2}{2}\right), \; h_2(x) = \exp\left(-x\right), \; h_3(x) = \left(1 + \frac{x^2}{2\alpha}\right)^{-\alpha}, \; h_4(x) = (1 + \sqrt{2\nu}x)e^{-\sqrt{2\nu}x},$$

respectively; where, $\alpha$ in $h_3$ is a positive-valued scale-mixture parameter, and the $\nu$ in $h_4$ is a parameter that controls the smoothness of the kernel results (Zhao et al., 2022; Genton, 2001).

One of the simplest kernel functions above is the Gaussian kernel, which is mostly used in machine learning problems and only depends on bandwidth parameter $\sigma$. The Gaussian kernel tends to 0 and 1 when $\sigma \to 0$ and $\sigma \to \infty$, respectively. Both situations lead to $\mathrm{MMD}^2$ being zero. Hence, the choice of the parameter

---

[11] A comprehensive study that explains why the RB ratio is a more appropriate measure of evidence than the BF can also be found in Al-Labadi et al. (2023).

$\sigma$ has a crucial effect on the performance of this kernel. Numerous methods are proposed to choose the value of $\sigma$, however, there is no definitive optimization method for this problem. The median heuristic is one of the first methods used in choosing $\sigma$ empirically and will be denoted in our experimental results by $\sigma_{MH}$. More precisely, for two samples $\{\mathbf{X}_i\}_{i=1}^n$ and $\{\mathbf{Y}_i\}_{i=1}^m$, the $\sigma_{MH}$ is considered as the median of $\{||\mathbf{X}_i - \mathbf{Y}_j||^2 : 1 \le i \le n, 1 \le j \le m\}$, which is mostly used in kernel-based tests (Schölkopf et al., 2002). Selecting $\sigma$ based on maximizing the power of two-sample problems is another strategy considered by Jitkrittum et al. (2016). The selection of the MMD bandwidth on held-out data to maximize power was first proposed by Gretton et al. (2012b) for linear-time estimates and by Sutherland et al. (2016) for quadratic-time estimates. Recently, bandwidth selection without data splitting has been proposed for quadratic (Schrab et al., 2021) and linear (Schrab et al., 2022) MMD estimates. Regarding the choice of $\sigma$ in kernel-based GANs, a common idea is assigning several fixed values to $\sigma$ and then considering the mixture of their corresponding Gaussian kernel. This strategy has received much attention and shown an acceptable performance in training GANs[12].

# E  Training Evaluation

## E.1  Traditional Approaches

Evaluating the quality of samples generated by GANs is considered to assess the mode collapse problem (Zhang, 2021).

### E.1.1  Fréchet Inception Distance (FID)

The FID is a widely used metric to assess the similarity between the distribution of generated images and real images (Heusel et al., 2017). It is based on the concept of comparing the statistics of feature representations of these images. Specifically, it computes the Fréchet distance between two multivariate Gaussian distributions fitted to the feature representations of the inception network for real and generated images.

Let $\{\phi_{\mathrm{Inc}}(\mathbf{X}_i)\}_{i=1}^n$ be the feature representations for the real images $\{\mathbf{X}_i\}_{i=1}^n$ in the inception network, if $\mu_{\phi_{\mathrm{Inc}}(\mathbf{X})}$ and $\Sigma_{\phi_{\mathrm{Inc}}(\mathbf{X})}$ are the sample mean vector and covariance matrix of $\{\phi_{\mathrm{Inc}}(\mathbf{X}_i)\}_{i=1}^n$, and $\mu_{\phi_{\mathrm{Inc}}(\mathbf{Y})}$ and $\Sigma_{\phi_{\mathrm{Inc}}(\mathbf{Y})}$ are the corresponding statistics for the generated images $\{\mathbf{Y}_i(\boldsymbol{\omega}^*)\}_{i=1}^n$, then the FID is defined as:

$$\mathrm{FID} = \|\mu_{\phi_{\mathrm{Inc}}(\mathbf{X})} - \mu_{\phi_{\mathrm{Inc}}(\mathbf{Y})}\|^2 + \mathrm{Tr}(\Sigma_{\phi_{\mathrm{Inc}}(\mathbf{X})} + \Sigma_{\phi_{\mathrm{Inc}}(\mathbf{Y})} - 2(\Sigma_{\phi_{\mathrm{Inc}}(\mathbf{X})}\Sigma_{\phi_{\mathrm{Inc}}(\mathbf{Y})})^{\frac{1}{2}}),$$

where "Tr" denotes matrix trace.

A lower FID indicates that the generated images are more similar to the real images.

### E.1.2  Kernel Inception Distance (KID)

The KID is another metric used to evaluate the quality of generated images. Unlike FID, which assumes the feature representations follow a Gaussian distribution, KID is based on the MMD between the real and generated images' feature representations (Bińkowski et al., 2018) as

$$\mathrm{KID} = \mathrm{MMD}^2(F_{\phi_{\mathrm{Inc}}(\mathbf{X}),n}, F_{\phi_{\mathrm{Inc}}(\mathbf{Y}),n}),$$

where $F_{\phi_{\mathrm{Inc}}(\mathbf{X}),n}$ denotes the empirical distribution corresponding to the sample $\{\phi_{\mathrm{Inc}}(\mathbf{X}_i)\}_{i=1}^n$.

The KID uses the polynomial kernel function in calculating MMD, which is defined as:

$$k(\phi_{\mathrm{Inc}}(\mathbf{X}_i), \phi_{\mathrm{Inc}}(\mathbf{Y}_i(\boldsymbol{\omega}^*))) = ((\phi_{\mathrm{Inc}}(\mathbf{X}_i))^\top \phi_{\mathrm{Inc}}(\mathbf{Y}_i(\boldsymbol{\omega}^*)) + 1)^\nu,$$

where $\nu$ is the degree of the polynomial and "T" denotes matrix transpose.

Similarly to FID, a lower KID indicates higher similarity between the real and generated images.

---

[12]For further details, see Li et al. (2015) and Li et al. (2017).

### E.2 An MMD Matching Score Function

To develop a stronger method for evaluating the differences between real and generated data manifolds, we propose using the MMD dissimilarity measure as follows: For $i = 1, \ldots, r_{mb}$, let $\{\mathbf{X}_{i_j}\}_{j=1}^{n_{mb}}$ and $\{\mathbf{Y}_{i_j}(\boldsymbol{\omega}^*)\}_{j=1}^{n_{mb}}$ be two samples drawn, respectively, from the real dataset $\mathbf{X}_1, \ldots, \mathbf{X}_n$ and the generated dataset $\mathbf{Y}_1(\boldsymbol{\omega}^*), \ldots,$ $\mathbf{Y}_n(\boldsymbol{\omega}^*)$ with the same sample size $n_{mb} < n$. Then, we define the MMD-based matching score as

$$\text{MMDS} = \max_{i \in \{1, \ldots, r_{mb}\}} \text{MMD}^2(F_{n_{mb}}(i), F_{G_{\boldsymbol{\omega}^*}, n_{mb}}(i)), \tag{24}$$

where, $\text{MMD}^2(F_{n_{mb}}(i), F_{G_{\boldsymbol{\omega}^*}, n_{mb}}(i))$ is the MMD approximation given by Equation (2, main paper) using samples $\{\mathbf{X}_{i_j}\}_{j=1}^{n_{mb}}$ and $\{\mathbf{Y}_{i_j}(\boldsymbol{\omega}^*)\}_{j=1}^{n_{mb}}$ (mini-batch samples). Our proposed matching score returns the maximum value of the MMD approximation between a subset of the real and a subset of the generated dataset with the same size $n_{mb}$ (mini-batch sample size) over $r_{mb}$ resamplings (mini-batch iteration). According to Equation (2, main paper), all components of mini-batch samples are compared together in the MMD measure, which provides a comprehensive assessment between subsets of the data in each iteration. Eventually, it is obvious smaller values of $MMDS$ indicate better quality and more diversity of the generated samples.

## F    Additional Experiments

### F.1    The Semi-BNP Test

To further illustrate the difference in performance between the BNP and FNP tests, we conducted tests on two alternative distributions: $F_1 = N(0, \sigma^2)$ for $\sigma^2 \in [1, 4]$ and $F_1 = 0.5N(-1 + \upsilon, 1) + 0.5N(1 - \upsilon, 1)$ for $\upsilon \in [0, 1]$. The corresponding results are reported in Figure 7 and 8 for univariate cases with $n = 50$. Figure 7(a) specifically shows that the proposed test exhibits a higher growth rate of the AUC when $\sigma^2$ is increased compared to the other tests. Additionally, Figure 7(b) indicates that our test starts to detect differences earlier than other tests ($\sigma^2 \geq 1.67$). Similar results can be found in Figure 8 for mixture distribution with various means.

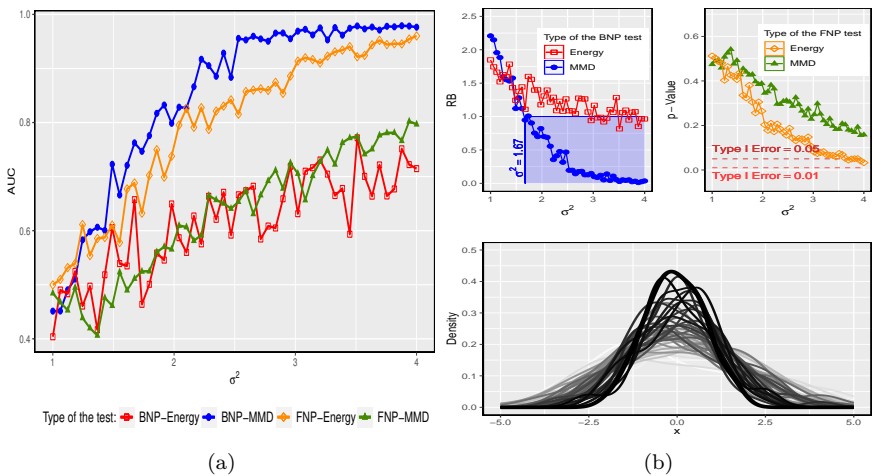

Figure 7: (a) AUC values in testing alternative $F_1 = N(0, \sigma^2)$ for $\sigma^2 \in (1, 4)$ in variance shift example. (b)-Top: Test critical values against different values of $\sigma^2$. (b)-Bottom: The lighter density corresponds to a larger value of $\sigma^2$.

Figure 9 provides a more focused comparison between the semi-BNP test and its Bayesian competitor, the BNP energy test. This figure illustrates the proportion of rejecting $\mathcal{H}_0$ over the 100 samples for both Bayesian tests mentioned, across different data dimensions. The first row of Figure 9 represents the type I error, while the remaining rows represent the test power. The figure demonstrates the effectiveness of the

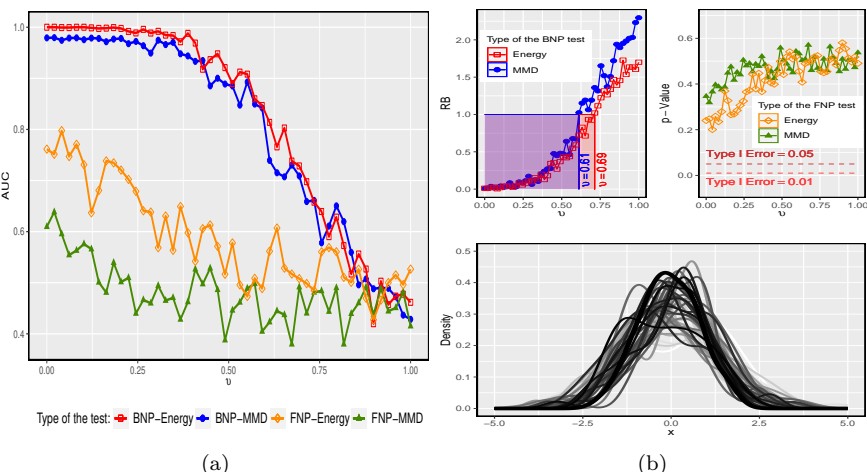

(a)

(b)

Figure 8: (a) AUC values in testing alternative $F_1 = 0.5N(-1 + v, 1) + 0.5N(1 - v, 1)$ for $v \in (0, 1)$ in mixture example. (b)-Top: Test critical values against different values of $\sigma^2$. (b)-Bottom: The lighter density corresponds to a smaller value of $v$.

semi-BNP kernel-based test in detecting differences, especially in scenarios involving variance shift, heavy tail, and kurtosis examples, where the BNP-energy test does not perform optimally in high sample sizes.

Moreover, to conduct a comprehensive analysis of the large sample property of all the tests in comparison, we present Table 2 for sample sizes $n = 500, 1000$. This table clearly demonstrates the weak performance of the BNP-Energy test in particular scenarios that are currently being mentioned.

Additionally, to display the asymptotic behavior of the posterior-based estimator compared to the baseline (3) (FNP-MMD), we provide the density plot of both under the null hypothesis in Figure 10 demonstrating the superior performance of the semi-BNP estimator in terms of faster convergence to zero.

### F.2 The Semi-BNP GAN

Now, we examine the performance of the proposed GAN through additional datasets, the details of which are given below. The generated samples are shown in Figures 11. Generally, the generated images using semi-BNP GAN show better resolution than the FNP GAN. The MMD scores presented in Table 3 are also evidence to demonstrate this claim. To further assess the performance of MMD-based GANs, we report the commonly used Fréchet inception distance (FID) and the Kernel inception distance (KID) metrics (Bińkowski et al., 2018). These metrics are well-suited for evaluating the performance of GANs. The corresponding scores[13] are reported in Table 3. Similar to our MMD scores, the smaller values of FID and KID show better performance of the GAN.

### F.2.1 Bone Marrow Biopsy Dataset (Tomczak & Welling, 2016):

The bone marrow biopsy (BMB) dataset is a collection of histopathology of BMB images corresponding to 16 patients with some types of blood cancer and anemia: 10 patients for training, 3 for testing, and 3 for validation. This dataset contains 10,800 images in the size of $28 \times 28$ pixels, 6,800 of which are considered for the training set. The rest of the images have been divided into two sets of equal size for testing and validation. The whole dataset can be found at `https://github.com/jmtomczak/vae_householder_flow/tree/master/datasets/histopathologyGray`. The results based on 6800 training images are presented in Figure 11-(a-c).

---

[13]The codes to compute the KID and FID are available at `https://github.com/mbinkowski/MMD-GAN/blob/master/gan/compute_scores.py`.

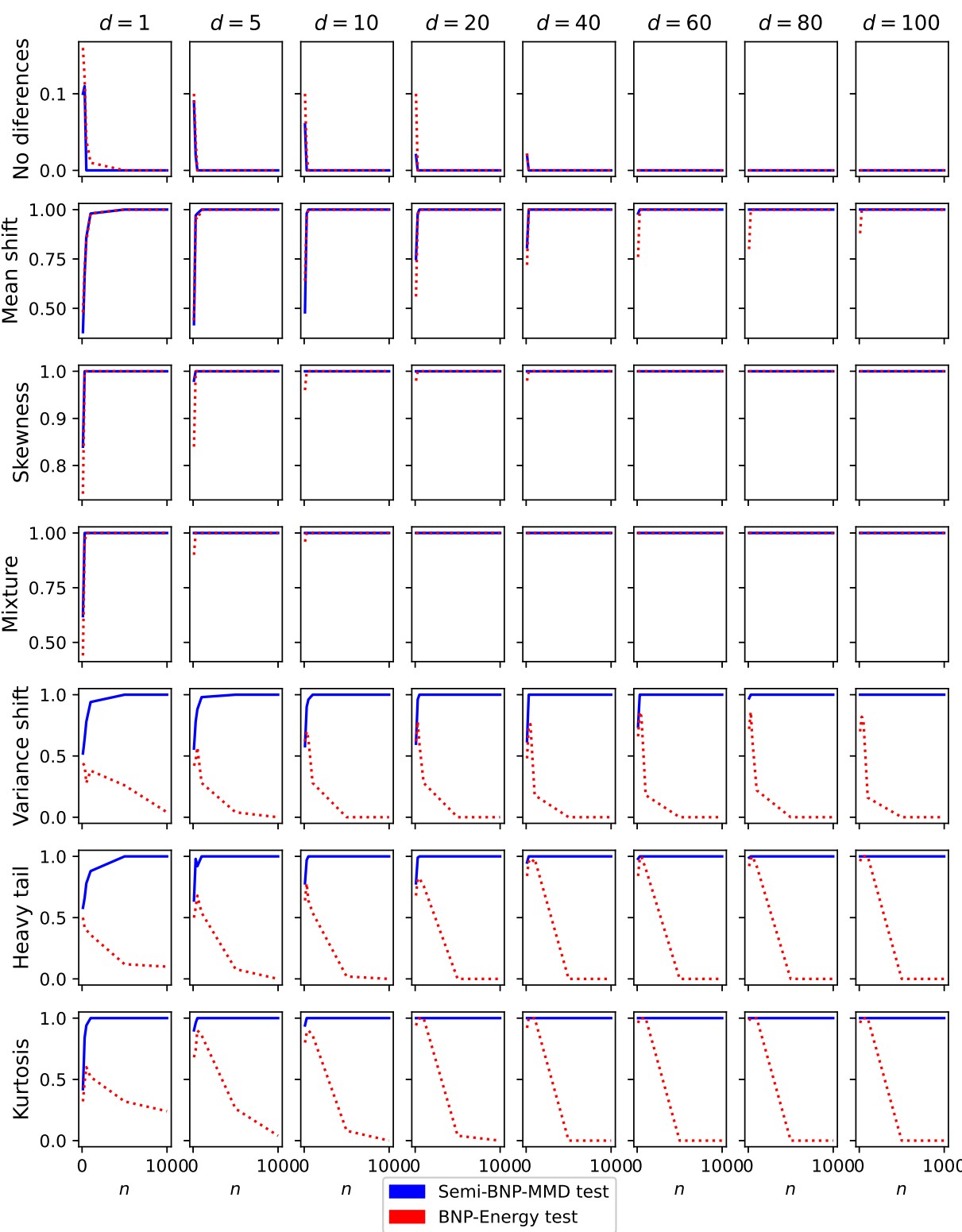

Figure 9: The proportion of rejecting $\mathcal{H}_0$ out of 100 replications against sample of sizes $n = 10, \ldots, 1000$ based on using $a = 25$, $\ell = 1000$, $\epsilon = 10^{-3}$ in equation 4, $M = 20$ for the semi-BNP-MMD (blue line) and BNP-energy (red dotted) tests.

Table 2: The average of RB, the average of its strength (Str ), and the relevant AUC out of 100 replications based on using $a = 25$, $\ell = 1000$, $\epsilon = 10^{-3}$ in equation 4, $M = 20$, and bandwidth parameter $\sigma = 80$ in RBF kernel for two sample of data with $n = 500, 1000$.

| | | BNP | | | | | | | | FNP | | | | | | | |
| | | MMD | | | | Energy | | | | MMD | | | | Energy | | | |
| | | RB(Str) | | AUC | | RB(Str) | | AUC | | P.value | | AUC | | P.value | | AUC | |
| Example | d | 500 | 1000 | 500 | 1000 | 500 | 1000 | 500 | 1000 | 500 | 1000 | 500 | 1000 | 500 | 1000 | 500 | 1000 |
|---|---|---|---|---|---|---|---|---|---|---|---|---|---|---|---|---|---|
| No diferences | 1 | 4.72(0.78) | 6.53(0.80) | | | 3.75(0.60) | 4.30(0.60) | | | 0.52 | 0.50 | | | 0.48 | 0.49 | | |
| | 5 | 18.84(0.86) | 19.65(0.93) | | | 18.74(0.88) | 19.58(0.76) | | | 0.50 | 0.51 | | | 0.51 | 0.44 | | |
| | 10 | 19.98(0.92) | 20(1) | | | 20(1) | 20(1) | | | 0.51 | 0.50 | | | 0.53 | 0.48 | | |
| | 20 | 20(1) | 20(1) | | | 20(1) | 20(1) | | | 0.53 | 0.51 | | | 0.51 | 0.44 | | |
| | 40 | 20(1) | 20(1) | | | 20(1) | 20(1) | | | 0.45 | 0.52 | | | 0.51 | 0.50 | | |
| | 60 | 20(1) | 20(1) | | | 20(1) | 20(1) | | | 0.51 | 0.50 | | | 0.50 | 0.53 | | |
| | 80 | 20(1) | 20(1) | | | 20(1) | 20(1) | | | 0.49 | 0.48 | | | 0.54 | 0.49 | | |
| | 100 | 20(1) | 20(1) | | | 20(1) | 20(1) | | | 0.49 | 0.48 | | | 0.51 | 0.50 | | |
| Mean shift | 1 | 0(0) | 0(0) | 1 | 1 | 0(0) | 0(0) | 0.98 | 0.98 | 0.001 | 0.001 | 1 | 1 | 0.004 | 0.004 | 1 | 1 |
| | 5 | 0(0) | 0(0) | 1 | 1 | 0(0) | 0(0) | 1 | 1 | 0.001 | 0.001 | 1 | 1 | 0.004 | 0.004 | 1 | 1 |
| | 10 | 0(0) | 0(0) | 1 | 1 | 0(0) | 0(0) | 1 | 1 | 0.001 | 0.001 | 1 | 1 | 0.004 | 0.004 | 1 | 1 |
| | 20 | 0(0) | 0(0) | 1 | 1 | 0(0) | 0(0) | 1 | 1 | 0.001 | 0.001 | 1 | 1 | 0.004 | 0.004 | 1 | 1 |
| | 40 | 0(0) | 0(0) | 1 | 1 | 0(0) | 0(0) | 1 | 1 | 0.001 | 0.001 | 1 | 1 | 0.004 | 0.004 | 1 | 1 |
| | 60 | 0(0) | 0(0) | 1 | 1 | 0(0) | 0(0) | 1 | 1 | 0.001 | 0.001 | 1 | 1 | 0.004 | 0.004 | 1 | 1 |
| | 80 | 0(0) | 0(0) | 1 | 1 | 0(0) | 0(0) | 1 | 1 | 0.001 | 0.001 | 1 | 1 | 0.004 | 0.004 | 1 | 1 |
| | 100 | 0(0) | 0(0) | 1 | 1 | 0(0) | 0(0) | 1 | 1 | 0.001 | 0.001 | 1 | 1 | 0.004 | 0.004 | 1 | 1 |
| Skewness | 1 | 0(0) | 0(0) | 1 | 1 | 0(0) | 0(0) | 1 | 1 | 0.001 | 0.001 | 1 | 1 | 0.004 | 0.004 | 1 | 1 |
| | 5 | 0(0) | 0(0) | 1 | 1 | 0(0) | 0(0) | 1 | 1 | 0.001 | 0.001 | 1 | 1 | 0.004 | 0.004 | 1 | 1 |
| | 10 | 0(0) | 0(0) | 1 | 1 | 0(0) | 0(0) | 1 | 1 | 0.001 | 0.001 | 1 | 1 | 0.004 | 0.004 | 1 | 1 |
| | 20 | 0(0) | 0(0) | 1 | 1 | 0(0) | 0(0) | 1 | 1 | 0.001 | 0.001 | 1 | 1 | 0.004 | 0.004 | 1 | 1 |
| | 40 | 0(0) | 0(0) | 1 | 1 | 0(0) | 0(0) | 1 | 1 | 0.001 | 0.001 | 1 | 1 | 0.004 | 0.004 | 1 | 1 |
| | 60 | 0(0) | 0(0) | 1 | 1 | 0(0) | 0(0) | 1 | 1 | 0.001 | 0.001 | 1 | 1 | 0.004 | 0.004 | 1 | 1 |
| | 80 | 0(0) | 0(0) | 1 | 1 | 0(0) | 0(0) | 1 | 1 | 0.001 | 0.001 | 1 | 1 | 0.004 | 0.004 | 1 | 1 |
| | 100 | 0(0) | 0(0) | 1 | 1 | 0(0) | 0(0) | 1 | 1 | 0.001 | 0.001 | 1 | 1 | 0.004 | 0.004 | 1 | 1 |
| Mixture | 1 | 0(0) | 0(0) | 1 | 1 | 0(0) | 0(0) | 1 | 1 | 0.06 | 0.01 | 0.93 | 0.99 | 0.004 | 0.004 | 1 | 1 |
| | 5 | 0(0) | 0(0) | 1 | 1 | 0(0) | 0(0) | 1 | 1 | 0.001 | 0.001 | 1 | 1 | 0.004 | 0.004 | 1 | 1 |
| | 10 | 0(0) | 0(0) | 1 | 1 | 0(0) | 0(0) | 1 | 1 | 0.001 | 0.001 | 1 | 1 | 0.004 | 0.004 | 1 | 1 |
| | 20 | 0(0) | 0(0) | 1 | 1 | 0(0) | 0(0) | 1 | 1 | 0.001 | 0.001 | 1 | 1 | 0.004 | 0.004 | 1 | 1 |
| | 40 | 0(0) | 0(0) | 1 | 1 | 0(0) | 0(0) | 1 | 1 | 0.001 | 0.001 | 1 | 1 | 0.004 | 0.004 | 1 | 1 |
| | 60 | 0(0) | 0(0) | 1 | 1 | 0(0) | 0(0) | 1 | 1 | 0.001 | 0.001 | 1 | 1 | 0.004 | 0.004 | 1 | 1 |
| | 80 | 0(0) | 0(0) | 1 | 1 | 0(0) | 0(0) | 1 | 1 | 0.001 | 0.001 | 1 | 1 | 0.004 | 0.004 | 1 | 1 |
| | 100 | 0(0) | 0(0) | 1 | 1 | 0(0) | 0(0) | 1 | 1 | 0.001 | 0.001 | 1 | 1 | 0.004 | 0.004 | 1 | 1 |
| Variance shift | 1 | 0.01(0) | 0(0) | 1 | 1 | 1.73(0.59) | 2.10(0.59) | 0.93 | 0.81 | 0.07 | 0.01 | 0.93 | 0.99 | 0.006 | 0.004 | 0.99 | 1 |
| | 5 | 0.42(0.07) | 0.40(0.08) | 0.99 | 1 | 4.42(0.72) | 7.30(0.70) | 0.73 | 0.64 | 0.001 | 0.001 | 1 | 1 | 0.004 | 0.004 | 1 | 1 |
| | 10 | 0.39(0.06) | 0.22(0.06) | 1 | 1 | 8.69(0.66) | 13.12(0.73) | 0.55 | 0.40 | 0.001 | 0.001 | 1 | 1 | 0.004 | 0.004 | 1 | 1 |
| | 20 | 0(0) | 0(0) | 1 | 1 | 13.43(0.78) | 18.12(0.69) | 0.35 | 0.07 | 0.001 | 0.001 | 1 | 1 | 0.004 | 0.004 | 1 | 1 |
| | 40 | 0(0) | 0(0) | 1 | 1 | 18.01(0.68) | 19.82(0.68) | 0.11 | 0 | 0.001 | 0.001 | 1 | 1 | 0.004 | 0.004 | 1 | 1 |
| | 60 | 0(0) | 0(0) | 1 | 1 | 19.19(0.55) | 19.98(0.94) | 0.02 | 0 | 0.001 | 0.001 | 1 | 1 | 0.004 | 0.004 | 1 | 1 |
| | 80 | 0(0) | 0(0) | 1 | 1 | 19.64(0.47) | 20(1) | 0 | 0 | 0.001 | 0.001 | 1 | 1 | 0.004 | 0.004 | 1 | 1 |
| | 100 | 0(0) | 0(0) | 1 | 1 | 19.82(0.64) | 20(1) | 0 | 0 | 0.001 | 0.001 | 1 | 1 | 0.004 | 0.004 | 1 | 1 |
| Heavy tail | 1 | 0.05(0) | 0(0) | 1 | 1 | 1.65(0.54) | 1.70(0.54) | 0.96 | 0.99 | 0.03 | 0.004 | 0.96 | 0.99 | 0.01 | 0.005 | 0.98 | 0.99 |
| | 5 | 0.04(0) | 0.02(0) | 1 | 1 | 2.89(0.71) | 4.53(0.74) | 0.91 | 0.76 | 0.001 | 0.001 | 1 | 1 | 0.004 | 0.004 | 1 | 1 |
| | 10 | 0(0) | 0(0) | 1 | 1 | 4.49(0.78) | 7.87(0.73) | 0.78 | 0.64 | 0.001 | 0.001 | 1 | 1 | 0.004 | 0.004 | 1 | 1 |
| | 20 | 0(0) | 0(0) | 1 | 1 | 5.66(0.76) | 11.73(0.75) | 0.77 | 0.42 | 0.001 | 0.001 | 1 | 1 | 0.004 | 0.004 | 1 | 1 |
| | 40 | 0(0) | 0(0) | 1 | 1 | 9.40(0.79) | 16.41(0.78) | 0.54 | 0.20 | 0.001 | 0.001 | 1 | 1 | 0.004 | 0.004 | 1 | 1 |
| | 60 | 0(0) | 0(0) | 1 | 1 | 11.02(0.74) | 18.06(0.82) | 0.52 | 0.16 | 0.001 | 0.001 | 1 | 1 | 0.004 | 0.004 | 1 | 1 |
| | 80 | 0(0) | 0(0) | 1 | 1 | 12.53(0.77) | 18.51(0.90) | 0.41 | 0.09 | 0.001 | 0.001 | 1 | 1 | 0.004 | 0.004 | 1 | 1 |
| | 100 | 0(0) | 0(0) | 1 | 1 | 13.17(0.75) | 19.07(0.97) | 0.30 | 0.06 | 0.001 | 0.001 | 1 | 1 | 0.004 | 0.004 | 1 | 1 |
| Kurtosis | 1 | 0(0) | 0(0) | 1 | 1 | 1.23(0.42) | 1.55(0.52) | 0.96 | 0.95 | 0.002 | 0.001 | 0.99 | 1 | 0.004 | 0.004 | 1 | 1 |
| | 5 | 0(0) | 0(0) | 1 | 1 | 1.75(0.59) | 3.54(0.70) | 0.96 | 0.88 | 0.001 | 0.001 | 1 | 1 | 0.004 | 0.004 | 1 | 1 |
| | 10 | 0(0) | 0(0) | 1 | 1 | 2.81(0.66) | 6.41(0.76) | 0.94 | 0.75 | 0.001 | 0.001 | 1 | 1 | 0.004 | 0.004 | 1 | 1 |
| | 20 | 0(0) | 0(0) | 1 | 1 | 4.63(0.71) | 9.90(0.78) | 0.84 | 0.51 | 0.001 | 0.001 | 1 | 1 | 0.004 | 0.004 | 1 | 1 |
| | 40 | 0(0) | 0(0) | 1 | 1 | 5.70(0.73) | 13.43(0.77) | 0.74 | 0.28 | 0.001 | 0.001 | 1 | 1 | 0.004 | 0.004 | 1 | 1 |
| | 60 | 0(0) | 0(0) | 1 | 1 | 7.06(0.75) | 16.38(0.81) | 0.72 | 0.23 | 0.001 | 0.001 | 1 | 1 | 0.004 | 0.004 | 1 | 1 |
| | 80 | 0(0) | 0(0) | 1 | 1 | 8.11(0.79) | 17.50(0.83) | 0.71 | 0.13 | 0.001 | 0.001 | 1 | 1 | 0.004 | 0.004 | 1 | 1 |
| | 100 | 0(0) | 0(0) | 1 | 1 | 8.83(0.78) | 18.52(0.89) | 0.55 | 0.09 | 0.001 | 0.001 | 1 | 1 | 0.004 | 0.004 | 1 | 1 |

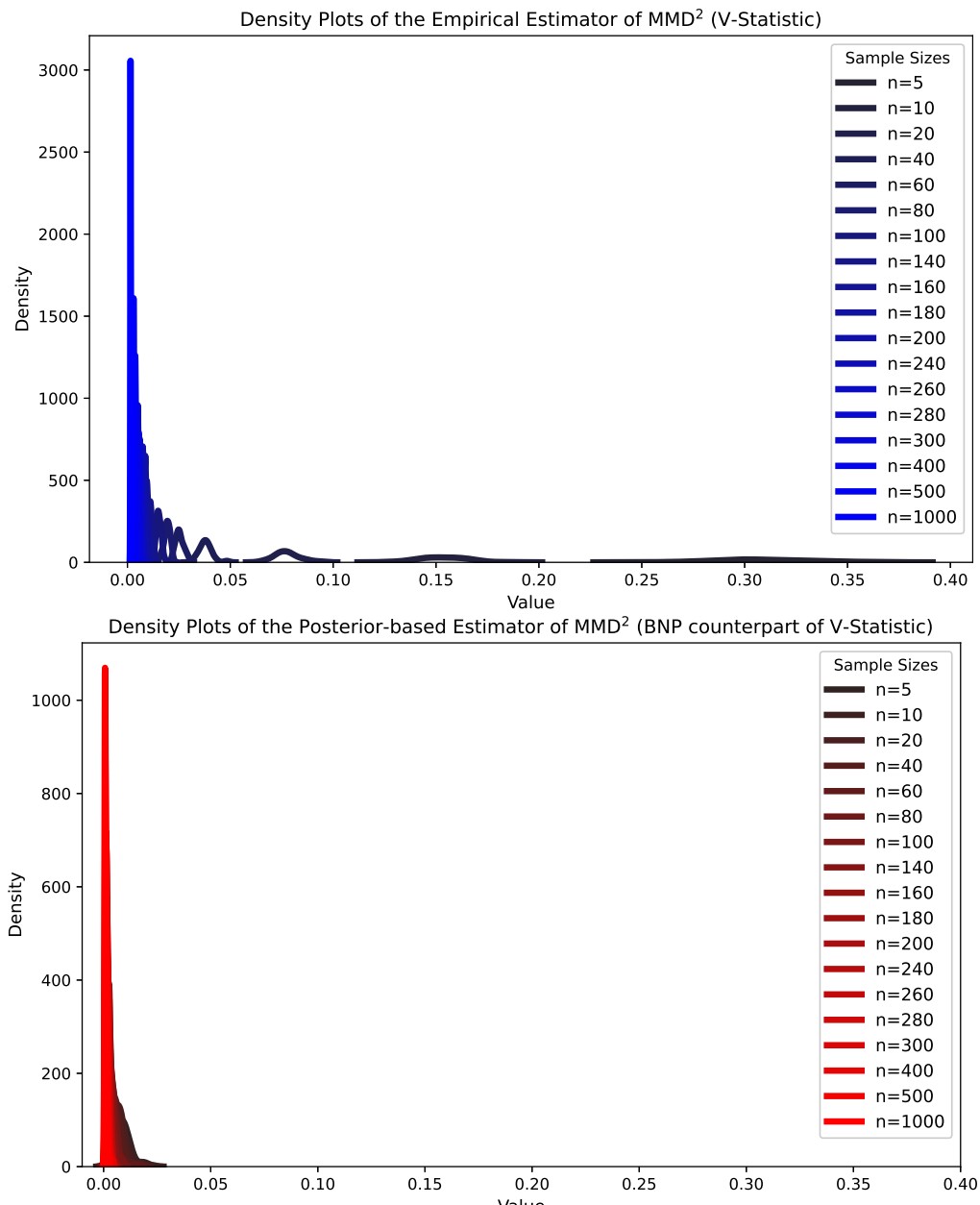

Figure 10: Density plots of the semi-BNP and FNP estimators of the MMD for various sample sizes.

### F.2.2 Labeled Faces in the Wild Dataset (Huang et al., 2008):

The labeled faces in the wild dataset (LFD) include 13,000 facial image samples with 1,024 ($32 \times 32$) dimensions. The dataset is available at `https://conradsanderson.id.au/lfwcrop/`.

### F.2.3 Brain Tumor MRI Dataset (Nickparvar, 2021):

In the last experiment, we consider a more challenging medical dataset including brain MRI images available at `https://www.kaggle.com/dsv/2645886`. This dataset has two groups including training and testing sets. Both are classified into four classes: glioma, meningioma, no tumor, and pituitary. To train the networks, we consider all 5,712 training images. The images vary in size and have extra margins. We use a pre-processing

code[14] to remove margins and then resize images to $50 \times 50$ pixels. We also scale the pixel value of prepared images to range 0-1 to make the range of distribution of feature values equal and prevent any errors in the backpropagation computation.

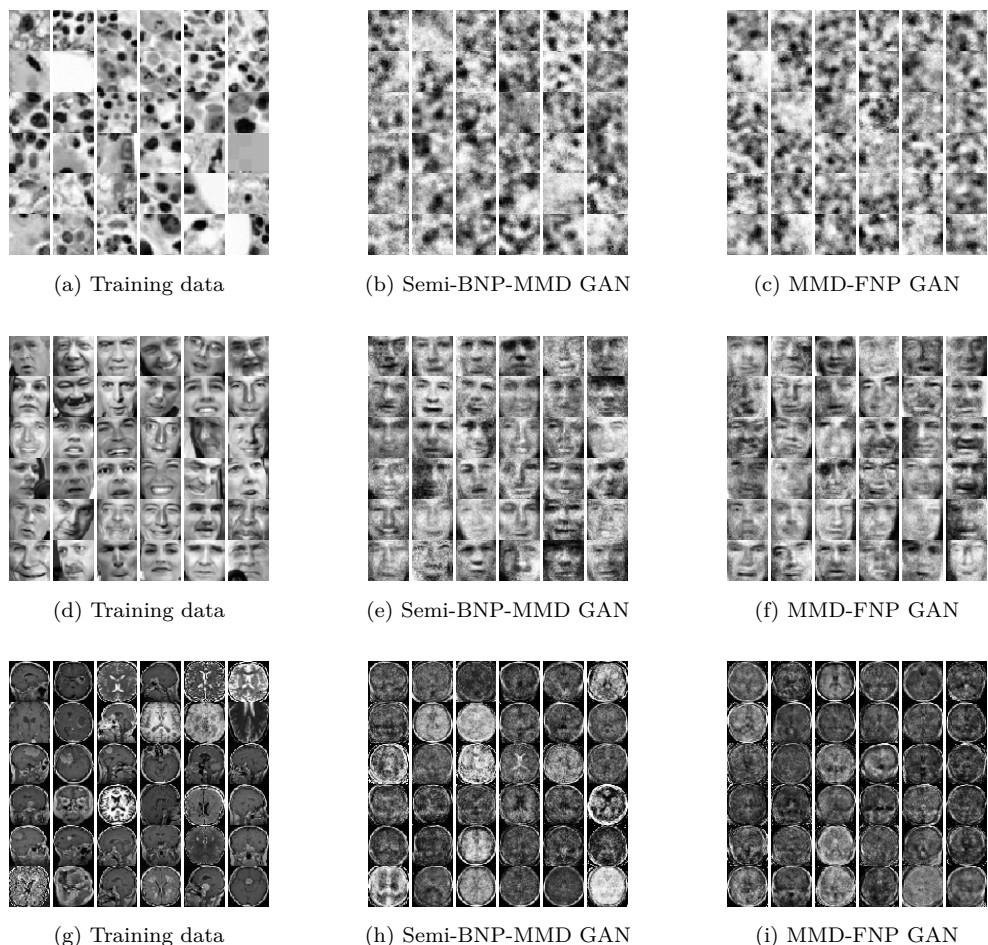

Figure 11: Generated samples of sizes $(6 \times 6)$ from semi-BNP-MMD and MMD-FNP GAN for the BMB and LFW datasets using a mixture of Gaussian kernels in 40,000 iterations.

Table 3: The values of MMD, KID, and FID scores for four groups of datasets considering $n_{mb} = 1000$ and $r_{mb} = 1000$ in equation 24.

| Scores | Dataset | | | | | | | |
|---|---|---|---|---|---|---|---|---|
| | MNIST | | BMB | | LFW | | MRI | |
| | Semi-BNP | FNP | Semi-BNP | FNP | Semi-BNP | FNP | Semi-BNP | FNP |
| MMD | 0.0384 | 0.0404 | 0.0285 | 0.0315 | 0.0281 | 0.0302 | 0.2059 | 0.2231 |
| KID | 0.0034 | 0.0046 | 0.0030 | 0.0036 | 0.0019 | 0.0026 | 0.0260 | 0.0264 |
| FID | 35.560 | 37.934 | 17.006 | 17.264 | 14.010 | 14.473 | 87.975 | 87.831 |

---

[14]https://github.com/masoudnick/Brain-Tumor-MRI-Classification/blob/main/Preprocessing.py

### F.2.4  Learning Rate Comparison: Semi-BNP-MMD GAN versus FNP Counterpart

To assess whether the proposed discriminator used in the Semi-BNP-MMD GAN leads to faster or better convergence of the generated samples compared to the baseline proposed by Li et al. (2015), we consider the synthetic distribution $\frac{1}{2}N(-\mathbf{1}_d, I_d) + \frac{1}{2}N(\mathbf{1}_d, I_d)$ as the true distribution and provide the corresponding MMD values for both models over 20,000 iterations in the data generation process, as shown in Figure 12. Our proposed GAN clearly displays a higher speed of convergence for the corresponding cost function to zero, and thus better performance compared to the baseline.

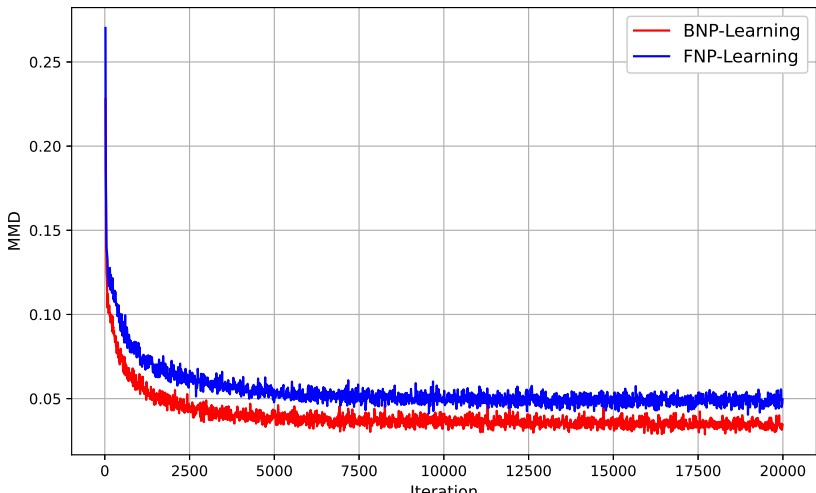

Figure 12: Learning rate: Values of the cost function in the proposed GAN and its frequentist counterpart (Li et al., 2015) over 20,000 iterations.

## G  More Discussion on the Potential Research

GANs are increasingly used in medical imaging applications which are effective tools for tasks such as medical imaging reconstructions. The synthetic images generated have often been proven to be valuable especially when the original image is noisy or expensive to obtain. GANs have also been used for generating images in cross-modality synthesis problems, where we observe magnetic resonance imaging (MRI) for a given patient but want to generate computed tomography (CT) images for that same patient (Wolterink et al., 2017). This type of generative method for medical imaging can drastically reduce the time and cost of obtaining data if the quality of the synthetic examples is sufficiently high. GANs have also been used in a diagnostic capacity–for example, in detecting brain lesions in images (Alex et al., 2017).

Here, the GAN is trained by distinguishing between labeled data of brain images that contain and do not contain lesions. Then, the discriminator of the GAN is used to detect brain lesions on new images. However, GANs are far less commonly used for tasks like diagnosis. According to a survey on medical imaging research in GANs, less than 10% of the top papers surveyed were dedicated towards making diagnoses, whereas the vast majority of papers were dedicated towards generating realistic synthetic examples of medical images for further analysis (Yi et al., 2019). We believe this is because where the cost of making errors in diagnosis is immediately consequential to people, unlike other AI applications where GANs are largely used.

We plan to extend the current work by mapping the data to a lower dimensional space using a variational auto-encoder (VAE), a dimensionality reduction model helps to reduce the noise in data and tries to optimize the cost function between the real data and fake data in the code space. There is a significant potential to combine the elements of Stein variational gradient descent (SVGD) and GANs. For instance, SVGD can be used for the inference of latent variables to approximate the variational distribution (the distribution of the latent variable given the observed dataset) in a VAE-GAN. Then, we will propose a 3D semi-BNP GAN in the code space to improve the ability of the GAN to generate medical datasets. The VAE method should

further reduce the chance of mode collapse and the 3D semi-BNP GAN will reduce the blurriness of the generated samples that may be caused by using the VAE. In future work, our model will be able to generate 3D images and, hence, increase the resolution of images, especially for MRI images. We hope that our future work will make an impact in the field of medical imaging.

## H    Notations

| Notation | Definition |
|---|---|
| $N(\cdot, \cdot)$ | Normal distribution |
| $LN(\cdot, \cdot)$ | Lognormal distribution |
| $t_3(\cdot, \cdot)$ | $t$-distribution with 3 degrees of freedom |
| $LG(\cdot, \cdot)$ | Logistic distribution |
| $B_d$ | $d \times d$ matrix with 0.25 on the main diagonal and 0.2 off the diagonal |
| $\mathbf{c}_d$ | $d$-dimensional column vector of $c$'s |
| $I_d$ | $d \times d$ identical matrix |

In all distribution notations, the first component represents the mean vector and the second component represents the covariance matrix.

## I    A Review on the Approximate Bayesian Computation

For a given data space $\mathfrak{X}$ and the set of Borel probability distributions $\mathcal{B}(\mathfrak{X})$, consider the observations $\mathbf{x}_{1:n}$ drawn from $F_{\text{True}} \in \mathcal{B}(\mathfrak{X})$. In the standard Bayesian perspective, given a parametric model $\mathcal{B}_\theta(\mathfrak{X}) = \{F_\theta : \theta \in \Theta\} \subset \mathcal{B}(\mathfrak{X})$, a prior $\pi$ is placed on the parameter space $\theta$. After observing data $\mathbf{x}_{1:n}$, the prior is updated to obtain a posterior distribution given by:

$$\pi(\theta|\mathbf{x}_{1:n}) = \frac{L(\mathbf{x}_{1:n}|\theta)\pi(\theta)}{\int_\Theta L(\mathbf{x}_{1:n}|\theta)\pi(\theta)d\theta}$$

where $L(\mathbf{x}_{1:n}|\theta) = \prod_{i=1}^n f_{\mathbf{X}_i}(\mathbf{x}_i|\theta)$ is the likelihood function and $f_\theta$ is the density function corresponding to the distribution $F_\theta$.

For the standard Bayesian inference to be considered well-specified, there must exist a parameter $\theta_0 \in \Theta$ such that the distribution $F_{\theta_0}$ matches the true data distribution $F_{\text{True}}$, i.e., $F_{\theta_0} = F_{\text{True}}$. When the parametric model $F_\Theta$ does not contain $F_{\text{True}}$ (i.e., there is no $\theta_0 \in \Theta$ such that $F_{\theta_0} = F_{\text{True}}$), the model is said to be misspecified. This can lead to problems in the standard Bayesian inference, as the posterior distribution will be based on an incorrect model assumption.

ABC addresses some of these challenges, particularly in scenarios where the likelihood function is intractable or difficult to compute. ABC circumvents the need for explicit likelihood evaluations by using simulations, summary statistics, and a comparison mechanism between simulated and observed data. Below shows how ABC works within this framework:

1. **Simulation**: Generate synthetic datasets by sampling parameters $\theta$ from the prior distribution $\pi$ and simulating data $\mathbf{y}_{1:n}$ from $F_\theta$. Here $F_\theta$ corresponds to the distribution of a generative model with implicitly defined likelihood functions.

2. **Summary Statistics**: Reduce both the observed data $\mathbf{x}_{1:n}$ and the simulated data $\mathbf{y}_{1:n}$ to lower-dimensional summary statistics $S(\mathbf{x}_{1:n})$ and $S(\mathbf{y}_{1:n})$. These summary statistics should capture the essential features of the data relevant to the parameters of interest.

3. **Comparison**: Compute a distance metric between $S(\mathbf{x}_{1:n})$ and $S(\mathbf{y}_{1:n})$.

4. **Acceptance Criterion**: Accept the parameter values $\theta$ if the distance between $S(\mathbf{x}_{1:n})$ and $S(\mathbf{y}_{1:n})$ is less than a predefined tolerance level $\epsilon$. This results in an approximate posterior distribution based on the accepted parameters.

5. **Posterior Approximation**: The ABC posterior distribution is then given by:

$$\pi_\epsilon(\theta|S(\mathbf{x}_{1:n})) \propto \int_{\mathfrak{X}} \cdots \int_{\mathfrak{X}} \prod_{i=1}^{n} \pi(\theta) f_{S(Y_i)}(S(y_i)|\theta) \mathbb{I}(\delta(S(\mathbf{x}_i), S(\mathbf{y}_i)) \leq \epsilon) \, dy_1 \cdots dy_n$$

where $\mathbb{I}$ is an indicator function that equals 1 if the distance $\delta$ is within the tolerance $\epsilon$, and 0 otherwise (Beaumont, 2019).

As a result, the ABC posterior distribution converges to the standard posterior as $\epsilon$ approaches 0 (Beaumont, 2019; Dellaporta et al., 2022). This convergence follows from the fact that as $\epsilon$ decreases, the ABC posterior places increasing weight on $\theta$ values that generate simulated data close to the observed data, effectively approximating the likelihood function. However, as already mentioned, the standard Bayesian posterior can be sensitive to model misspecification. If the assumed model is not a good representation of the true data-generating process, the Bayesian posterior (and thus the ABC posterior when $\epsilon \to 0$) may give misleading inferences. This lack of robustness to model misspecification can lead to poor performance in practice.

