# OpenReview forum: "A Semi-Bayesian Nonparametric Estimator of the Maximum Mean Discrepancy Measure: Applications in Goodness-of-Fit Testing and Generative Adversarial Networks"
_TMLR — Accepted by TMLR_

### Review · Reviewer_46Pm · 2024-05-19

**Summary Of Contributions:**

This manuscript proposed a semi-nonparametric estimator for maximum mean discrepancy that can be applied to model with intractable likelihood and unknown distributional form. The authors further show how to perform goodness-of-fit test and generative modeling based on the proposed estimator. Empirical results are provided to demonstrate the effectiveness of the proposed method.

**Audience:**

Yes

**Broader Impact Concerns:**

N/A.

**Claims And Evidence:**

No

**Requested Changes:**

Please completely revise Section 4 as the weakness part suggests.

**Strengths And Weaknesses:**

### Strengths:
* As a Bayesian non-parametric people the high level idea is easy to follow.

### Weaknesses:
* The notation is in a chaos. For example, $H$ is used to denote both base measure and the Gamma random variables. Some terms are used without definition, e.g. $H_N$ and $mmd$ in Eq (8). I suggest the authors make a proofreading and fix the issues like this. Furthermore, I also suggest to use the same font style to define the samples to make the presentation more clear. Now $x$ is also used to denote the samples and make me confused at the first glance.
* The words in Section 4 are hard to understand. Why not using a flow chart?
* Several claim in Section 4 is confusing. How can we choose $H=F_2$? If not, the reasoning in the middle of Page 6 is nonsense.
* For the discussion on how to set $a$, the claims on two statistics tend to $0$ is only correct when the hypothesis holds, then why we cannot make any decision on the hypothesis? Definitely when $a$ is extremely large then prior and posterior has so little difference which may make the testing statistics non-informative, but this does not make the reasoning here sound.
* The authors mentioned there should be a consistency result for the estimator (9) and (10), but I don’t find it. Also (7) should not converge to $M/i_0$ as it does not depend on these terms. So the claim here is not so rigorous.
* It is not clear why the proposed new test will overcome the frequentist counterpart. At least the authors should show the (asymptotic) distribution of the testing statistics and compare with the frequentist counterpart. (Note that for MMD we have lots of the tools from U-/V- statistics to characterize it).
* For the goodness-of-fit testing, how do you set the criterion for the proposed test?
* For the MNIST experiments, I would like to see some quantitative evaluation like e.g. FID scores to compare.

---

> ### Author Response · Authors · 2024-07-31
> **Response to the reviewer 46Pm**
>
> We thank the reviewer for their constructive suggestions and comments! We will make sure to address the issues pointed out by the reviewer during revision. We will consider the structural changes suggested by the reviewer to improve the comprehensibility section 4. We address the reviewer’s questions below.
>
> > 1. The notation ...
>
> Thank you for pointing this out. We have changed the Gamma random variables to be denoted as $\lbrace\Gamma_{i,N}\rbrace_{i=1}^{N}$. We have also defined $H_N$ in Theorem 1 as the ECDF corresponding to the base measure $H$.
> Additionally, we have revised the sample notations to keep them as $\mathbf{X}\_{1:n}$ when they are random variables, and modified to the lowercase $\mathbf{x}\_{1:n}$ when we refer to the observations of random variables.
>
> > 2. The words ...
>
> Thanks for your suggestion. We have added a flow chart to illustrate the test procedure, which can be found in Figure 6 in the Appendix. We have also referenced this diagram on page 7 to help clarify the content of Section 4.
>
> >3.  Several claim ...
>
> Choosing the base measure $H$ as the hypothesized model is the basic component of the BNP test using belief ratio (see Al-Labadi \& Evans, 2018; Al-Labadi et al., 2021a;b; Al-Labadi et al., 2022a;b). By choosing $H=F_2$, we are considering the semi-BNP-MMD statistic under the null hypothesis to provide the threshold value $d_{i/M}$ that the prior-based estimator and the posterior-based estimator are compared against (we have mentioned this in the first paragraph after Equation (12)). We have also modified the sentence on page 6 as follows to further clarify the reason:
>
> - Here, supporting $\mathcal{H}_0$ by
> $\pi\_{\mathrm{MMD}^2}(\cdot)$ means to place most of the prior mass on zero. To enforce this term on $ \pi\_{\mathrm{MMD}^2}(\cdot) $, it is enough to set $H=F_2$ in $DP(a, H)$, which is deduced from the Theorem 1, part (iii), **to make the denominator of (8) concentrate most of its mass around zero; otherwise, any other choice for $H$ contradicts the RB ratio rule in GOFs.**
>
> >4. For the discussion ...
>
> Thank you for your question. However, we did not claim that the two statistics tend to 0 only when the hypothesis holds. We stated that, following Corollary 3(i), the expectation of both statistics tends to 0 as $a$ tends to infinity, iff $H=F_2$ (base measure $H$ is chosen as the hypothesized model). We missed adding **if and only if $H=F_2$** at the end of this sentence: **\emph{Corollaries 3(i) also clearly point to this issue in the informative prior case, as both expectations of $\mathrm{MMD}^{2}\_{\mathrm{BNP}}(F\_{1,N}^{pos},F_{2,m})$ and $\mathrm{MMD}^{2}\_{\mathrm{BNP}}(F\_{1,N}^{pri},F\_{2,m})$ tend to $0$ as $a\rightarrow\infty$, $N\rightarrow\infty$, and $m\rightarrow\infty$},** as we supposed $H=F\_2$, as mentioned in the previous paragraph.
> We have now added this clarification. Additionally, Figure 6 visually clarifies the rule for such a choice for $H$.
>
> We recommend choosing $a$ less than half of the sample size to avoid the excessive effect of the prior $H$ on the test results (informative prior). Generally, the informative prior, which leads to both PDFs of the statistics coinciding, is shown in Figure 6 when $a$ is chosen to be much larger than the sample size ($a \gg n$). Note that in the case $H=F_2$, this coincidence will be massed around 0.
>
> >5. The authors ...
>
> Thank you for bringing this issue to our attention. We have completely revised this part of the paper. Instead of discussing the convergence of Equation (8) (Equation (7) in the previous version), we now focus on the convergence of Equations (10) and (11) (Equations (9) and (10) in the previous version). We have added a detailed explanation after Equation (12) to clarify our meaning regarding the consistency of $RB$ and $Str$.
>
> >6. It is not clear ...
>
> Thank you for pointing that out.
> We should mention that the baseline here is the V-statistic, which is considered as the discriminator used in Li et al. (2015) (our frequentist nonparametric (FNP) counterpart in the GAN).
> To address your concern, we have provided the asymptotic behavior of the PDF of both semi-BNP-MMD and the baseline by including Figure 10. We have provided an explanation and referenced this figure in the last paragraph of Section 3.
>
> >7. For the goodness-of-fit testing, ...
>
> We have responded to this comment by adding the following sentence in the paragraph after Equation (12):
>
> - Here $d_{i/M}$ is the threshold value that the prior-based estimator (the semi-BNP statistic under the null hypothesis) and the posterior-based estimator is compared against.
>
> >8. For the MNIST experiments, ...
>
> We already provided these results in Table 3, columns 2 and 3.

---

### Review · Reviewer_Rb2H · 2024-05-24

**Summary Of Contributions:**

1. The paper introduces a semi-Bayesian nonparametric (semi-BNP) estimator for the Maximum Mean Discrepancy (MMD) measure and demonstrates that it can be used to create an improved goodness-of-fit test compared to frequentist and other BNP methods.

2. The proposed semi-BNP estimator is applied as a discriminator to Generative Adversarial Networks (GANs) where it shows comparable performance with other MMD-based approaches.

3. The paper provides theoretical properties of the proposed estimator and discusses its robustness and consistency.

**Audience:**

Yes

**Claims And Evidence:**

Yes

**Requested Changes:**

1. In Section 2.2. can you provide an example of summary statistics and how they are used by ABC to estimate the model parameters.

2. Can you clarify why the ABC tends to approximate the standard Bayesian posterior as the threshold approaches zero and why that would be a problem?

3. Can you also give an example of the exponential of robust loss functions used in Generalized Bayesian Inference?

4. How is this approach connected to sample based variational inference approaches like SVGD [1-3]?

5. Why is the best test performance in Fig 1 always achieved by setting the bandwidth parameter to 80 despite the range of distributions considered?

6. Can the GAN experiments be augmented by considering data from a synthetic/simulated distribution and showing that the proposed discriminator leads to faster/better convergence of the generator samples compared to baselines?

[1] Liu, Qiang, Jason Lee, and Michael Jordan. "A kernelized Stein discrepancy for goodness-of-fit tests." International conference on machine learning. PMLR, 2016.

[2] Liu, Qiang, and Dilin Wang. "Stein variational gradient descent: A general purpose bayesian inference algorithm." Advances in neural information processing systems 29 (2016).

[3] Feng, Yihao, Dilin Wang, and Qiang Liu. "Learning to draw samples with amortized stein variational gradient descent." arXiv preprint arXiv:1707.06626 (2017).

**Strengths And Weaknesses:**

Strengths:

1. The proposed approach can be applied to hypothesis testing with intractable statistical models where the theoretical analysis and its empirical improvements over baselines make it a suitable alternative
2. The use of the estimator to design a novel discriminator for GANs along with the accompanying empirical evaluation shows that the idea is also applicable to high dimensional practical settings where goodness-of-fit estimates are required.

Weaknesses:

1. The exposition of the background in Section 2.2. is a bit rushed and may overwhelm readers unfamiliar with the literature (see requested changes below)
2. The results of the GAN experiments in Fig 3 appear inconclusive but that is a common issue with GAN evaluation, so I do not see it as a major concern.

---

> ### Author Response · Authors · 2024-07-31
> **Response to the reviewer Rb2H**
>
> We thank the reviewer for their positive feedback on our paper, and for their constructive feedback. Responding to the requested changes in order:
>
> > 1.
>
> Thank you for your comment. In the first paragraph of the Section 2.2, we explain how a summary statistic can be employed in the ABC procedures.
>
> Now, we have clarified the steps:
>
>  - In ABC, we first sample a proposal by sampling from a prior distribution placed on the parameter space of the generative model **(Step 1)**. Rather than inferring parameters directly from the posterior distribution, we compare the summary statistics of the simulated data, given the current state of the parameters, with those of the observed data using a discrepancy measure **(Step 2)**. The simulated parameter values corresponding to the accepted summary statistics are retained if the distance falls within a predetermined threshold **(step 3)**.
>
> On the other hand, choosing summary statistics in such problems depends on the application in question. We mention two examples from population genetics and modified Section 2.2 to say:
>
>
>
> - Identifying informative summary statistics in ABC is a challenging task **and it depends on the specific application of the data being analyzed, such as the mean effective heterozygosity or the mean of variance in repeat numbers in genetic populations  (Blum \& Francois, 2010; Csillery et al., 2012).**
>
>
> In addition to the changes made above, we have also added Section I in the Appendix to include further explanation on ABC.
>
> > 2.
>
>  A comprehensive discussion on these two questions is directly provided by Dellaporta et al. (Section 2.2, 2022) and Beaumont (2019). To save space in the main paper, we refer the reader to those references. However, to address these questions, we have included a discussion in Section I of the Appendix to clarify the details.
>
> Briefly, this convergence follows from the fact that as the threshold decreases, the ABC posterior places increasing weight on the model's parameters that generate simulated data close to the observed data, effectively approximating the likelihood function. However, the standard Bayesian posterior can be sensitive to model misspecification. If the assumed model is not a good representation of the true data-generating process, the Bayesian posterior (and thus the ABC posterior when the threshold tends to 0) may give a misleading inference. This lack of robustness to model misspecification can lead to poor performance in practice.
>
> > 3.
>
> The following sentence has been added to Section 2.2:
>
> - For a detailed exploration, Jewson et al. (2018) offers various examples, including the use of exponential of the Hellinger divergence within the GBI.
>
> >4.
>
> Thank you for bringing our attention to this related area of work. We have incorporated the three references you cited into the introduction and Section 2.2 to say:
>
> - Stein variational gradient descent (Liu \& Wang, 2016) is another variational inference method that uses gradient-based updates derived from Stein’s method and kernel functions. It leverages Bayesian principles to iteratively move particles to match the posterior distribution by minimizing Kullback-Leibler divergence, particularly when exact inference is computationally intractable (Feng et al., 2017).
>
> Additionally, in the section "More Discussion on the Potential Research," Section G of the Appendix, we have briefly discussed how SVGD can be connected to our procedure as follows:
>
> - There is a significant potential to combine the elements of Stein variational gradient descent (SVGD) and GANs. For instance, SVGD can be used for the inference of latent variables to approximate the variational distribution (the distribution of the latent variable given the observed dataset) in a VAE-GAN.
>
> >5.
>
> Numerous methods have been proposed in the literature to choose the value of $\sigma$, yet there is no single definitive optimization method for this problem. In our study, we numerically selected the bandwidth parameter to maximize the area under the receiver operating characteristic curve (AUC). We evaluated on a range of values: 2, 5, 10, 20, 40, 80, and applied the median heuristic for estimating this parameter. Notably, we found that $\sigma=80$ consistently yielded an AUC of 1 across all simulated examples. While this numerical result suggests $\sigma=80$ as optimal in our tests,  other values of $\sigma$ may achieve optimal AUC values
>
> >6.
>
> Thank you for your suggestion. We have addressed this by providing a plot (Figure 12 in Appendix F.2.4) that compares the learning rates of our proposed procedure with the baseline. The baseline model, from Li et al. (2015), uses the V-statistic version of the MMD. This figure demonstrates a faster convergence rate in our semi-BNP GAN compared to the baseline.

---

### Review · Reviewer_ZKRi · 2024-07-16

**Summary Of Contributions:**

This paper proposes a semi-parametric Bayesian nonparametric (semi-BNP) estimator for approximating minimum mean discrepancy (MMD) between two probability distributions.
The asymptotic properties of the proposed estimator are presented in theorems and corollary.
The authors develop a goodness of fit test with the relative belief (RB) factor between the densities of the proposed posterior- and the prior-based estimators.
The authors apply the proposed estimator also to generative adversarial network (GAN) learning. The generator’s properties, generalization error and robustness, are shown in lemmas.
The semi-BNP test and the semi-BNP GAN are investigated with synthetic datasets and the MNIST dataset, respectively.

**Audience:**

Yes

**Broader Impact Concerns:**

I have no concern on the ethical implications of the work.

**Claims And Evidence:**

No

**Requested Changes:**

Some adjustments on the following three points are critical.

p.6: The discussion right after Eq.(7) says that the proposed test is based on the property of Theorem 1 (iii). However, this property is given in the form of an inequality, which is not so tight as it is derived for example with $1/m < 1$ (right after Eq.(15) of Appendix). I wonder how an accurate test can be constructed based on the inequality without assessing its tightness.

p.11, the third paragraph of Section 6.1: Although it is said that the best performance, AUC=1, is achieved for the bandwidth parameter 80, I wonder if AUC=1 is possible.
Doesn’t it mean that the type I error is not controlled appropriately? Since 80 is the largest bandwidth parameter tested and reported in the experiment, if it performed best, I wonder why the bandwidth parameters greater than 80 are not considered.

p.26: Although Appendix E discusses quality measures of generated samples by GANs, they are not used and reported in the experimental investigation section.

minor comments:
p.2, the last line: continues -> continuous
p.3, l.14 from the bottom: ``Monte Carlo Markov Chain’’ -> ``Markov Chain Monte Carlo’’
p.6, l.14 from the bottom: Corollaries -> Corollary
p.7, l.12: The phrase ``In the sense that’’ is confusing.
p.9, Lemma 5, the definition of $h(N,m,K,\epsilon)$: $N$ should be $n$. Also, in the assertion i, the function $h$ lacks the third argument, $K$.
p.20, l.1: Chebyshev’s inequality
p.25, l.3 from the bottom: As $\sigma \rightarrow $\infty$, does the Gaussian kernel tend to the linear kernel (not only it tends to 1)?
p.27, l.3: Some & (and) symbols are garbled.

**Strengths And Weaknesses:**

The proposed estimator for MMD is theoretically sound and its performance is promising. In particular, the theoretical results on the semi-BNP GAN in lemmas 4 and 5 are interesting.
However, there are some unclear discussions, especially in the experimental investigation section.

---

> ### Author Response · Authors · 2024-07-31
> **Response to the reviewer ZKRi**
>
> We sincerely thank the reviewer for recognizing the positive impact of our results and for providing thoughtful feedback. In the following response, we have carefully addressed all the concerns raised.
>
> >1.
>
> Thank you for your comment. You are correct. We have provided a discussion in the paragraph around Equation (10) to help readers understand the general core of the procedure by comparing the posterior to the prior. We also addressed your concern by including the following explanation at the end of the second paragraph after Equation (8) (Equation (7) in the previous version of the paper):
>
> - However, the above discussion provides a general understanding rather than a tight inequality for comparison around zero. For accuracy, we use a critical value $d_{i_{0}/M}$ and the interval $[0, d_{i_{0}/M})$ to approximate the RB, as addressed in (10).
>
> >2.
>
> Frequentist hypothesis tests are basically constructed based on controlling Type I errors (the probability of rejecting a true $\mathcal{H}\_0$). In Bayesian hypothesis testing, the focus shifts from controlling specific error rates to evaluating the evidence for or against a hypothesis based on updating the probability of the hypotheses given the data.
> There is no direct counterparts to Type I error in the Bayesian test. If the evidence (relative belief) incorrectly favors the alternative hypothesis $\mathcal{H}\_1$ over the null hypothesis $\mathcal{H}\_0$, it could be seen as analogous to a Type I error. However, this is not about a fixed error rate but rather the frequency with which the ratio of the posterior probability given the data to the prior is less than 1 for a true null hypothesis over a number of replicated experiments.
>
> We have mentioned this property of Bayesian tests at the end line of the second paragraph of Appendix C (Relative Belief Ratio: A Bayesian Measure of Evidence) as **In this scenario where evidence for the null hypothesis is plausible, the frequentist notion of controlling the probability of falsely rejecting $\mathcal{H}\_0$ (type I error) does not apply.**
>
> On the other hand, for the experiments given in Figure 1, the corresponding RB values are almost 0 for all $r=100$ alternative experiments (see Table 1 for $n=50$ and $d=60$). By considering $L=400$ in Algorithm 3 in the Appendix, we have $T=[0,0.05,0.1,\ldots,19.95,20]$ and $TPR=[\frac{0}{0+100},\frac{100}{0+100},\ldots,\frac{100}{0+100},\frac{100}{0+100}]$, resulting in an immediate jump from 0 to 1 in ROC curves and $AUC=1$. Additionally, since the RB values are quite large when $\mathcal{H}_0$ is true, many of the initial components of the $FPR$ vector will be 0.
>
> To address your concerns, we have updated Figure 1 to include ROC curves for $10^2, 10^4, 10^6$ to assess test performance as this parameter increases, and added the following explanation at the end of the relevant paragraph of Section 6.1:
>
> - For this $\sigma$, the AUC jumps from zero to 1 as either $\mathcal{H}_0$ is strongly rejected by too small values of RB (near to 0, the minimum RB) in alternative experiments or $\mathcal{H}_0$ is strongly accepted by large values of RB (near to 20, the maximum RB) in null experiments. As noted in the Appendix, if $\sigma$ is too small or too large, the MMD approaches zero, resulting in poor performance, as shown in Figure 1.
>
> >3.
>
> We have modified Appendix E by replacing these approaches with definition of commonly used Fr\'{e}chet inception distance (FID) and the Kernel inception distance (KID) metrics that we have reported in Table 3 of the Appendix.
>
> >4.
>
> We have fixed these errors. Regarding your question "*As $\sigma \rightarrow \infty$, does the Gaussian kernel tend to the linear kernel (not only it tends to 1)?*", we would like to clarify that the Gaussian kernel is defined as:
>
> \begin{equation}
> K(\mathbf{X}, \mathbf{Y}) = \exp\left(-\frac{\||\mathbf{X} - \mathbf{Y}\||^2}{2\sigma^2}\right)
> \end{equation}
> while the linear kernel is defined as:
> \begin{equation}
> K(\mathbf{X}, \mathbf{Y}) = \mathbf{X}^T \cdot \mathbf{Y},
> \end{equation}
> which is simply the dot product of the two vectors $\mathbf{X}$ and $\mathbf{Y}$. As $\sigma \to \infty$, the Gaussian kernel function tends to 1 regardless of the inputs $\mathbf{X}$ and $\mathbf{Y}$.

---

> > ### Comment · Reviewer_ZKRi · 2024-08-09
> > **limiting behavior of Gaussian kernel**
> >
> > Thank you for answering my questions. Regarding the minor comment on the limit of the Gaussian kernel, I meant if the behavior of the method, the semi BNP test in the case of this paper, approaches that with the linear kernel. For example, kernelized linear classfiers with the Gaussian kernel tend to yield linear boundaries in the limit $\sigma \rightarrow \infty$. It might be nicer to discuss whether such a limiting behavior also applies to MMD or not.

---

> ### Author Response · Authors · 2024-08-13
> **limiting behavior of Gaussian kernel in semi-BNP test**
>
> Thank you for your suggestion.
>
> For the semi-BNP test in this paper, this situation leads to a failure in judging the null hypothesis, as both the prior- and posterior-based MMD estimators approach zero, causing the RB value to approach 1. We already showed the weak performance for such situation by Figure 1. To address your concern, we will add the following sentence at the end of the first paragraph on page 12 in the next update of the paper:
>
> - In the context of this paper's semi-BNP test, both the prior- and posterior-based MMD estimators converge to zero under these conditions, causing their corresponding density functions to coincide at zero, resulting in $RB\approx 1$ and rendering the test unable to evaluate $\mathcal{H}_0$.

---

### Author Response · Authors · 2024-08-02

The authors would like to thank the reviewers for their helpful comments and we have written a careful rebuttal to the questions posed. Now, we invite any further questions that the reviewers may have regarding our paper.

---

### Decision · Action_Editor_R8cY · 2024-08-26

**Recommendation:** Accept as is

**Comment:**

I am happy to recommend acceptance of the paper.  The authors may consider addressing some of the lingering issues that reviewers had.  In particular, one of the reviewers felt the notation was still hard to follow ("the notation is complicated and it can be hard for the unfamiliar reader to follow") and, as mentioned above, there are some concerns about the amount of empirical evaluation in the paper.  However, I think the paper is solid enough that I will not insist on a revision to be further reviewed.

**Audience:**

Yes, this is definitely relevant to the TMLR audience.  Goodness-of-fit tests are relevant, as are GAN models, to the TMLR audience.

**Claims And Evidence:**

There are both theoretical and empirical claims made in the paper.  Reviewers were, on the whole, happy with these.  All three reviewers noted that the theoretical contributions are solid (multiple reviewers mentioned in their reviews or recommendations that the work was theoretically "sound").  On the empirical side, the reviewers appreciated the revision, which seemed to answer lingering questions.  One reviewer did note "regarding the evaluation of the semi BNP GAN, while quite a major revision has been made in the appendix, it is unclear this can be considered as a technical contribution;" another reviewer mentioned that "despite the novelty and apparent technical soundness of the approach, the empirical improvements seem a bit limited in significance".